# Post-Treatment of Synthetic Polyphenolic 1,3,4 Oxadiazole Compound A3, Attenuated Ischemic Stroke-Induced Neuroinflammation and Neurodegeneration

**DOI:** 10.3390/biom10060816

**Published:** 2020-05-26

**Authors:** Arooj Mohsin Alvi, Lina Tariq Al Kury, Muhammad Umar Ijaz, Fawad Ali Shah, Muhammad Tariq Khan, Ahmed Sadiq Sheikh, Humaira Nadeem, Arif-ullah Khan, Alam Zeb, Shupeng Li

**Affiliations:** 1Riphah Institute of Pharmaceutical Sciences, Riphah International University, Islamabad 44000, Pakistan; aroojalvi@hotmail.com (A.M.A.); tariq.khan@cust.edu.pk (M.T.K.); ahmedsadiq@gmail.com (A.S.S.); humaira.nadeem@riphah.edu.pk (H.N.); Arif.ullah@riphah.edu.pk (A.-u.K.); alam.zeb@riphah.edu.pk (A.Z.); 2State Key Laboratory of Oncogenomics, School of Chemical Biology and Biotechnology, Shenzhen Graduate School, Peking University, Shenzhen 518055, China; 3College of Natural and Health Sciences, Zayed University, Abu Dhabi 144534, UAE; Lina.AlKury@zu.ac.ae; 4Department of Zoology, Wildlife, and Fisheries, University of Agriculture, Faisalabad 38000, Pakistan; umar12ejaz@hotmail.com; 5Department of Pharmacy, Capital University of Science and Technology, Islamabad 44000, Pakistan

**Keywords:** A3, middle cerebral artery occlusion, all-trans retinoic acid, neurodegeneration, antioxidant system

## Abstract

Ischemic stroke is categorized by either permanent or transient blood flow obstruction, impeding the distribution of oxygen and essential nutrients to the brain. In this study, we examined the neuroprotective effects of compound A3, a synthetic polyphenolic drug product, against ischemic brain injury by employing an animal model of permanent middle cerebral artery occlusion (p-MCAO). Ischemic stroke induced significant elevation in the levels of reactive oxygen species and, ultimately, provoked inflammatory cascade. Here, we demonstrated that A3 upregulated the endogenous antioxidant enzymes, such as glutathione s-transferase (GST), glutathione (GSH), and reversed the ischemic-stroke-induced nitric oxide (NO) and lipid peroxidation (LPO) elevation in the peri-infarct cortical and striatal tissue, through the activation of endogenous antioxidant nuclear factor E2-related factor or nuclear factor erythroid 2 (Nrf2). In addition, A3 attenuated neuroinflammatory markers such as ionized calcium-binding adapter molecule-1 (Iba-1), cyclooxygenase-2 (COX-2), tumor necrotic factor-α (TNF-α), toll-like receptors (TLR4), and nuclear factor-κB (NF-κB) by down-regulating p-JNK as evidenced by immunohistochemical results. Moreover, treatment with A3 reduced the infarction area and neurobehavioral deficits. We employed ATRA to antagonize Nrf2, which abrogated the neuroprotective effects of A3 to further assess the possible involvement of the Nrf2 pathway, as demonstrated by increased infarction and hyperexpression of inflammatory markers. Together, our findings suggested that A3 could activate Nrf2, which in turn regulates the downstream antioxidants, eventually mitigating MCAO-induced neuroinflammation and neurodegeneration.

## 1. Introduction

Stroke is the most prominent cause of human disability and it might either be classified as transient or permanent, and it occurs as a result of occlusion in an artery irrigating the brain. Ischemic stroke has an 80% higher incidence as compared to other types of stroke and the middle cerebral artery (MCA) is the most common place for the occurrence of an ischemic stroke [1]. The prevention and treatment of stroke are of utmost importance with numerous medications under investigation. However, until now, no treatment strategy has proved to be effective by controlled clinical trials, except for the only Food and Drug Administration (FDA) approved recombinant tissue plasminogen activator (tPA), which acts through vascular recanalization and tends to have numerous limitations, of which, the transformation from ischemia to hemorrhage and narrow therapeutic window are the most prominent ones [2]. The post-stroke pathological processes can be categorized into acute (minutes to hours), subacute (hours to days), and chronic (days to months) phase [3,4]. The acute phase is initiated by a robust release of reactive oxygen species (ROS) and pro-inflammatory mediators immediately after the stroke onset, ultimately leading to the subacute and chronic phase, all contributing to neurodegeneration [5,6]. The molecular mechanisms underlying ischemic stroke injury are quite complex, primarily due to the involvement of numerous pathways that are associated with energy metabolism, oxidative stress, excitatory amino acid toxicity, inflammation, free radicals formation, calcium overload, and mitochondrial dysfunction, all of which lead to varying degrees of cellular damage [7]. After a transient occlusion, irreversible events, starting from ionic imbalance to subsequent membrane depolarization and calcium overload, are initiated, triggering oxidative stress and neuronal cell death [8]. ROS, like free radicals damage lipids, proteins, DNA, and cause intracellular organelle destruction via plasma and organelle membrane peroxidation [9]. Moreover, ROS production leads to the activation of downstream inflammatory and apoptotic cascading pathways that are associated with glia cells stimulation and the infiltration of granulocytes into the brain [9]. Numerous researches suggest that neurons in the penumbral region undergo apoptosis several hours or days after ischemia and could be rescued [10]. Thus, this region is salvageable and targeting this region to prevent apoptosis seems to be a rational approach to limit cerebral infarct following clinical stroke.

Nrf2 (nuclear factor E2-related factor or nuclear factor erythroid 2 or p45-related factor 2) has been extensively studied as a potential neuroprotective signaling pathway [11]. It is predominantly associated with Kelch-like ECH-associated protein 1 (keap1) in the cytoplasm as an inactive dimer. Keap1 suppresses Nrf2 activation and prevents its nuclear translocation. Upon exposure to stress signal, the complex is dissociated and Nrf2 is set free for nuclear translocation, where it triggers the innate antioxidant machinery of the cell. In the nucleus, Nrf2 binds to antioxidant response elements (AREs) and triggers transcription machinery of endogenous protective enzymes, including antioxidant genes, phase II detoxifying enzyme, anti-inflammatory genes, and molecular chaperones, and it further induces several cellular protective functions in numerous inflammatory reactions, malignant tumors, cardiovascular diseases, and respiratory diseases [12]. Hence, it might act as a prominent therapeutic target in acute ischemic stroke, owing to its imperative role in cellular resistance to exogenous toxic substances and oxidative stress.

All trans-retinoic acid (ATRA) is a vitamin-A derivative and binds to retinoic acid receptors (RARs) and demonstrates immunoregulatory activities in numerous models [13]. Conflicting activities have been attributed to ATRA, including some neuroprotective effects [14,15]. However, researchers have also documented the inhibitory effects of ATRA on the Nrf2-ARE pathway, where it either augmented Nrf2-Keap dimer in the cytoplasm or inhibited the activation of ARE [16,17]. In another study, the inhibitory effects of ATRA were reported, owing to retinoic acid receptors (RAR), ultimately inhibiting ARE driven genes [18]. It was employed as an Nrf2-inhibitor in the present study owing to its Nrf2 inhibitory potential.

Oxadiazoles are five-membered heterocyclic compounds presenting diverse biological activities. Previously, these agents have been extensively researched for various biological activities, including anti-inflammatory and antioxidant potentials [19,20,21]. These oxadiazoles can be considered as neuroinflammatory modulators owing to their high antioxidant potential. A novel 1,3,4 oxadiazole derivative (*N*-{4-[(5-sulfanyl-1,3,4-oxadiazol-2yl) methoxy] phenyl} acetamide (named as A3) was chosen based on its previously reported toxicity assessment, tumor inhibition, antioxidant, analgesic, and anti-inflammatory potential [20] (Figure 1). The previous study by Faheem et al. reported its relative safety profile using brine shrimp lethality assay (lethal concentration LC_50_ = 5 µg/mL), and acute toxicity studies in mice, which proved its safety up to a dose of 750 mg/kg. Furthermore, its cyclooxygenase-2 (COX-2) affinity through computational analysis was correlated with its analgesic and anti-inflammatory potential [20]. Its free radical scavenging activity and relative safety profile can be set as a foundation in order to further assess its in-vivo antioxidative potential. The present study aims to investigate and explore the neuroprotective effects of A3 in a rat model of permanent middle cerebral artery occlusion (p-MCAO). We will demonstrate whether A3 exerts its neuroprotective role in ischemic stroke via the Nrf2 dependent pathway. The outcome of this study will not only facilitate in interpreting the cascading mechanisms leading to inflammatory processes, but also provide insight into the use of Nrf2 as a therapeutic target in ischemic stroke.

## 2. Materials and Methods

### 2.1. Chemicals and Reagents

Proteinase K and PBS tablets were obtained from MP Biomedicals (Irvine, CA, USA). Hydrogen peroxide (H_2_O_2_), formaldehyde, trichloroacetic acid (TCA), reduced glutathione (GSH), *N*-(1-Naphthyl) ethylenediamine dihydrochloride, 5,5′-dithiobis-(2-nitro benzoic acid) (DTNB), and 1-chlor-2,4-dinitrobenzene (CDNB) were purchased from Sigma–Aldrich (St. Louis, MO, USA). Mouse monoclonal anti-*p*-NF-κB (SC-271908), mouse monoclonal anti-p-JNK (SC-6254), mouse monoclonal anti-TNF-α (SC-52B83), mouse monoclonal anti-COX-2 (SC-514489), rabbit polyclonal anti-Nrf2 (SC-722), mouse monoclonal anti-TLR4 (SC-293072), mouse anti-Iba-1 (sc-32725), and mouse monoclonal anti-HO-1 (SC-13691), along with Avidin-biotin complex (ABC kit, SC-516216) and 3,3′-Diaminobenzidine (DAB, SC-216567), were obtained from Santa Cruz Biotechnology (Dallas, TX, USA). The horseradish peroxidase-conjugated secondary antibodies (ab-6789, ab-6721) and mounting media (ab-10431) were purchased from Abcam UK. *p*-NF-κB ELISA kit (Cat # SU-B28069), Nrf2 ELISA kit (Cat # SU-B30429), HO-1 ELISA kit (Cat # SH-032529), and TNF-α ELISA kit (Cat # SU-B3098) were purchased from Shanghai Yuchun Biotechnology (Shanghai, China). ATRA (All trans-retinoic acid) of the highest analytical grade (99% HPLC) was purchased from the local pharmaceutical industry (GlaxoSmithKline, Islamabad, Pakistan).

### 2.2. Experimental Groups and Drug Treatment

In this study, adult male Sprague–Dawley rats weighing 230–260 g, 7–10 weeks were used, which were obtained from the local breeding facility of Riphah International University. They were kept in a standard animal room at 18–22 °C under circadian light and dark conditions with access to food and water ad libitum. All of the experimental protocols were recommended and approved from the research and the ethical committee (REC) of the Riphah Institute of Pharmaceutical Sciences (RIPS) (Approval ID: Ref. No. REC/RIPS/2018/14 and Date of Approval: 15 November 2018), and strictly adhered to the approved protocols, in addition, to ARRIVE guidelines with few exceptions [22,23]. We did not apply human endpoints for euthanizing the rats as the permanent MCAO model (24 h) is the most stressful invasive procedure and in which limited mobility with severe suffering is an established documented protocol, and our group was more interested in rats that survive this period. By this, we did not euthanize any rats until 24 h of the ischemic period. We applied all of the laboratory procedures to minimize rat sufferings, such as heating pad, sterilization, and fluid replenishment with normal saline. The rats were randomly divided into six experimental groups (Figure 2). The sham-operated control group (*n* = 17), vehicle-treated p-MCAO group (*n* = 17), A3 treated group both at low (5 mg/kg) (*n* = 17), and high dose (*n* = 16) (10 mg/kg), ATRA treated group (*n* = 15) and A3 + ATRA treated group (*n* = 15).

A3 and ATRA were both dissolved in saline (containing 5% DMSO) and all rats received a single intraperitoneal dose of either A3 (5 mg/kg or 10 mg/kg) or vehicle, 30 min after permanent MCAO or ATRA (5 mg/kg) 30 min before MCAO occlusion. All of the animals were decapitated 24 h after permanent MCAO and brain tissues were collected. A total of seven animals died during the experimental procedures, of which three were from the p-MCAO group, one from high dose A3, two from low dose A3, and one from the sham group, which were further adjusted by supplementing more animals. The reported reason for this mortality is edema formation, BBB leakage, and hypothalamic shutdown [24]. The ethics committee is mostly aware of the mortality in experimental setup, particularly in this model, as we constantly engaged them for our work.

### 2.3. Animal Preparation and MCAO Surgery

The animals were anesthetized with an I/P administration of a cocktail of xylazine (9 mg/kg) and ketamine (91 mg/kg). The body core temperature was maintained while using a heating pad. MCAO was carried out, as described previously [25,26,27,28]. Briefly, the right common carotid artery (R-CCA) and its bifurcating branches; internal and external carotid arteries were exposed after a midline cervical incision. Superior thyroid artery and the occipital artery, which are small protrusions from the external carotid artery, were identified and knotted with a thin black (6/0) silk and eventually ligated to allow free movement of the external carotid artery. The external common carotid artery was then tied with silk (6/0) near the hyoid bone that was located above the ligated superior thyroid artery and immediately incised near the bifurcating point. Throughout this procedure, extra care was exercised in order to avoid excessive bleeding. For occlusion, a blue nylon filament (3/0) with a blunted round end was used, which was inserted through the opening of the external carotid artery and pushed into the internal carotid artery up to 18–19 mm (depending upon the weight and age of the rat) until the origin of the middle cerebral artery (MCA), where a light resistance to the advancement of nylon indicated MCA occlusion. The nylon was then tied in place with the lumen of the external carotid artery and the skin was then sealed. All of the animals were subjected to carbon dioxide (CO_2_) euthanasia for tissue collection. The sham group underwent the whole procedure without nylon insertion. The only shortcoming in this method was the absence of the Doppler effect and relative blood flow measurement, though we constantly performed occlusion with suitable experimental skills. The exclusion criteria included animals showing no depressed signs or alteration in movements after awakening from anesthesia. No significant adverse effects were observed in drug-treated animals, as pilot toxicity studies indicated a significantly greater dose.

### 2.4. Neurobehavioral Tests

The neurobehavioral analysis was conducted after 24 h of occlusion in order to assess sensory-motor deficits. To ensure accuracy, tests were performed by an experienced, blinded observer. The Bederson scale and 28-point neuro scores were run in parallel to validate behavioral deficits and neuronal damage.

#### 2.4.1. Bederson Scale

The Bederson scale was assessed to test the neurological state of rats using previously published protocols [29]. Parameters, such as forelimb flexion, circling behavior, and resistance to lateral push, were assessed and grading was done on a scale of 0–3, with 0 representing no neurological deficit and 3 representing maximum neurological deficit.

#### 2.4.2. 28-Point Neuro Score

28-point neuro score was used to estimate the extent of sensorimotor deficits based on previously mentioned protocols [30]. Several tests were performed, including (i) circling, (ii) motility, (iii) general physical condition, (iv) righting reflex; ability to turn itself when placed on its back, (v) paw placement on the tabletop, (vi) its ability to climb itself up on a horizontal bar, (vii) ascending on an inclined platform, (viii) grip-strength, (ix) contralateral reflex, (x) contralateral rotation where it rotates when it is held by the base of its tail, and (xi) visual forepaw reaching. An aggregate score of 28 indicates a healthy neurological function, whereas 0 indicates severe neurological deficits. All of the experiments were performed following the SOPs that were approved by Stanford Behavioral and Functional Neuroscience Laboratory.

### 2.5. Determination of Oxidative Stress Markers

Oxidative stress markers, such as the glutathione s-transferase (GST) activity and reduced glutathione (GSH) levels, were determined to assess the degree of oxidative damage and the relative effect of the test drug. The brain tissue samples were homogenized in 0.1 M sodium phosphate buffer (PBS) (pH 7.4), which contained phenylmethylsulfonyl fluoride (PMSF) as a protease inhibitor, centrifuged at 4000× *g* for 10 min at 4 °C and supernatant was collected. GSH level was determined using a previously reported method with slight modifications [31]. 0.6 mM DTNB was dissolved in 0.2 M sodium phosphate, and 2 mL of this solution was mixed with 0.2 mL of the supernatant. Further, 0.2 M phosphate buffer was added to this mixture, to make a final volume of 3 mL. The absorbance of the reaction mixture was measured after 10 min at 412 nm while using a spectrophotometer. Phosphate buffer was used as blank, whereas the DTNB solution was used as control. Real absorbance was calculated by subtracting the absorbance of control from that of the tissue lysate. The final GSH values were expressed in µmoles/mg of proteins. Likely, for the determination of GST activity, the previously reported protocol was followed [32]. 5 mM GSH and 1 mM CDNB in 0.1 M phosphate buffer solution were freshly prepared. Three replicates of 1.2 mL reaction mixture were placed in a glass vial. Tissue supernatant (60 μL) was then added to the reaction mixture. Three blanks were also made while using 60 μL water instead of tissue lysate. Aliquots of 210 μL from the reaction mixture were pipetted in a microtiter plate and the reaction rate was recorded at 340 nm for 5 min at 23 °C using ELISA microplate reader (BioTek ELx808, Winooski, VT, USA). The GST activity was calculated using the extinction coefficient of the product formed and expressed as µmoles of CDNB conjugate/min/mg of protein.

### 2.6. LPO Assay

Lipid peroxidation assay was carried out by measuring thiobarbituric acid reactive substances (TBARS) using colorimetry, as described previously with slight modifications [33]. The assay mixture consists of 580 µL of 0.1 M phosphate buffer (pH 7.4), 200 µL supernatant, 20 µL of ferric chloride, 200 µL of 100 mM ascorbic acid and incubated on a water bath at 37 °C for 60 min. After one hour of incubation, the reaction was stopped by adding 1000 µL of 10% trichloroacetic acid (TCA) and 1000 µL of 0.66% thiobarbituric acid (TBA) to samples. The tubes were retained on the water bath for 20 min, cooled on an ice bath, and finally centrifuged at 3000× *g* for 10 min. Supernatant’s absorbance and blank (containing all reagents except test drug) were measured at 535 nm to determine the concentration of TBARS-nM/min/mg protein.

### 2.7. NO Assay

Nitric oxide assay was performed according to reported protocols with slight modifications [34]. 50 μL of tissue supernatant and 50 μL of normal saline were mixed with an equal volume of Griess reagent containing 1% sulfanilamide in 0.1% naphthyl ethylenediamine dihydrochloride, and 5% phosphoric acid in distilled water. The resultant reaction mixture was incubated at 37 °C for 30 min. The absorbance of the reaction mixture was detected at 546 nm while using ELISA microplate reader (BioTek ELx808) using standard sodium nitrite solution in order to calibrate the absorbance coefficient.

### 2.8. Brain Water Content

Brain edema was quantitated using the previously described method [35]. Briefly, the animals were decapitated under anesthesia and the brain was removed. The whole-brain was weighed immediately on an electronic analytical balance to obtain the wet weight. The brain tissues were then dried at 120 °C for 6–8 h to obtain the dry weight. The formula used was as follows:(Wet weight − Dry weight)/wet weight × 100

### 2.9. 2,3,4-Triphenyl Tetrazolium Chloride Staining

The rats were sacrificed under anesthesia, and brain tissue was collected. Three-millimeter slices were made from the frontal lobe using a sharp blade. These were then incubated at 37 °C for 20 min in a freshly prepared TTC solution (2% in PBS), in a water bath until a clear white/red coloring demarcation was observed. The slices were fixed using a 4% paraformaldehyde solution and then photographed for percent infarct area determination. ImageJ software (ImageJ 1.3; https://imagej.nih.gov/ij/) was used for infarct area determination and it was expressed as a percentage to the total area. To overcome the false reading of brain edema, the corrected brain infarction was calculated using the formula:Corrected infarct area = [left hemisphere area − (right hemisphere area − infarct area)]/100

These sections were then processed for paraffin blocks preparation using an embedding machine and rotary microtome (*n* = 7/group).

### 2.10. Hematoxylin Eosin Staining

After de-paraffinizing tissue slides using absolute xylene (100%), it was followed by rehydrating with absolute ethanol, gradient ethanolic concentrations (95% to 70%), and subsequently with distilled water. The slides were then rinsed with PBS and kept in hematoxylin for a total of 10 min. After due time, the slides were placed for 5 min under running tap water in a glass jar. Slides were then probed under the microscope for nuclear staining, and if staining was not clear, hematoxylin timing was increased. The slides were then treated with 1% HCl and 1% ammonia water for a short interval and immediately rinsed with water again. These were then immersed in an eosin solution for 5–10 min, followed by rinsing with water and finally air-dried. Slides were then dehydrated using graded ethanol (70%, 95%, and 100%), fixed in xylene and cover-slipped. The images were captured using an Olympus light microscope (Olympus, Tokyo, Japan) and analyzed using ImageJ software. Five images per group were captured and analyzed while focusing on neuronal shape, size, infiltrated cells, and vacuolation.

### 2.11. Immunohistochemical Analysis

Immunohistochemical analysis was performed, as described previously with minor modifications [28,36,37]. The tissue slides were subjected to deparaffinization protocol, which started from three different xylene treatments for 10 min. and then rehydrated in graded alcohol preparation (commencing from 100% to 70%, each wash for 5–10 min). These slides were subsequently rinsed with distilled water to clear ethanol remaining. The slides were processed for antigen retrieval with proteinase K and washed with 0.1 M PBS. The peroxidase activity was quenched by applying a diluted hydrogen peroxide solution (3% in methanol). After washing with 0.1 M PBS, slides were incubated with 5% NGS (normal goat serum) containing 0.1% Triton X-100 for a minimum of 1 h in a humidified chamber. After blocking, the slides were kept for overnight incubation at 4 °C with primary antibodies as anti-COX-2, anti-p-JNK, anti-TNF-α, anti-*p*-NF-κB, anti-Nrf2, anti-TLR4, anti-Iba-1, and anti-HO-1 (Dilution 1:100, Santa Cruz Biotechnology, Dallas, TX, USA). The next morning, after washing twice with 0.1 M PBS, they were incubated for 90 min. with biotinylated secondary antibodies (dilution factor 1:50) in a humidified chamber. The slides were again washed and incubated for 1 h with ABC reagents (Santa Cruz Biotechnology, USA) in a humidified chamber. Slides were then stained in DAB solution, washed with distilled water, dehydrated in graded ethanol, fixed in xylene, and cover-slipped using mounting medium. Immunohistochemical TIF (Tagged Image Format file) images of the slides were taken using a light microscope (Olympus, Tokyo, Japan) taking three images per slide. The number of immune-positive cells expressing p-JNK, TNF-α, *p*-NF-κB, COX-2, Nrf2, TLR4, Iba-1, and HO-1 in cortex/striatum/total area of the brain were counted through ImageJ software by first optimizing the TIF image background according to the threshold intensity and then analyzing the intensity for the number of immune-positive cells at the same threshold intensity for all groups. These were then expressed in terms of the relative integrated density for the number of immune-positive cells of the samples’ comparative to the sham.

### 2.12. Immunofluorescence Analysis

Immunofluorescence was performed, as described previously [28]. The slides were first deparaffinized with xylene and rehydrated with a series of ethanol concentrations (100%, 90%, 80%, and 70%) followed by washing with distilled water and PBS. Antigen retrieval was performed with protein kinase, and it was washed with 0.01 M PBS, followed by 1 h incubation in blocking solution containing 0.3% Triton X-100 and 3% normal serum in 0.01 M PBS, according to the source of a secondary antibody. After blocking, the slides were then incubated overnight at 4 °C, with a primary antibody against Nrf2 (1:100, Santa Cruz Biotechnology, Inc.). The next day, after PBS washing, the slides were treated with fluorescent-labeled secondary antibody tetramethylrhodamine isothiocyanate (TRITC, 1:50, Santa Cruz Biotechnology Inc.) for signal amplification, incubated for one hour in a dark chamber at room temperature, and then cover-slipped using Ultra Cruz mounting medium containing 4′,6-diamidino-2-phenylindole (DAPI) (Santa Cruz Biotechnology, Inc.). The TIF Images were then captured using a confocal scanning microscope (Flouview FV 1000, Olympus, Tokyo, Japan). Three images per group were taken and the fluorescence intensity of the same regions of cortex/total area of the TIF images for all groups was measured while using ImageJ software. The background of all the TIF images was optimized according to the threshold intensity and intensity of immunofluorescence was also analyzed at the same threshold intensity for all groups and the results were expressed as the relative density of sample relative to sham.

### 2.13. Enzyme-Linked Immunosorbent Assay (ELISA)

Rat Nrf2, *p*-NF-κB, HO-1, and TNF-α ELISA kits were used to quantify the expression of the respective proteins. Expression was measured as per manufacturer’s instructions (Shanghai Yuchun Biotechnology, Shanghai, China). An appropriate amount of tissues (50 mg) were first homogenized while using Silent Crusher M (Heidolph) at 15,000 RPM in PBS (2500 μL) containing PMSF as protease inhibitor and then centrifuged at 4000× *g* for 10 min and the supernatant was separated. The total protein concentration in each group was determined by the BCA method (Elabscience), and the equivalent quantity of protein was then loaded to determine the protein expression of Nrf2, *p*-NF-κB, TNF-α, and HO-1 while using ELISA microplate reader (BioTek ELx808) and the concentration (pg/mL) were then normalized to total protein content (pg/mg total protein).

### 2.14. Statistical Analysis

All of the data are presented as mean ± SEM. 2,3,4-triphenyl Tetrazolium Chloride Staining, neurobehavior results, and oxidative stress data were analyzed using One-Way ANOVA, followed by posthoc Bonferroni’s multiple comparison test using GraphPad Prism-8 software. For the rest of the data, Two-Way ANOVA followed by posthoc Bonferroni’s Multiple Comparison test was performed. ImageJ software (ImageJ 1.30; https://imagej.nih.gov/ij/) was used to analyze the morphological data. Symbols # or * represents significant difference values *p* < 0.05, ## or ** represent *p* < 0.01 and ### or *** represents *p* < 0.001 values for significant differences. Symbols * and # shows significant difference relative to sham and p-MCAO, respectively.

## 3. Results

### 3.1. Effects of Post-Treatment Dosage Regimen on Brain Infarction and Neuronal Cell Loss

The neuroprotective effect of A3 on acute neurological deficits was demonstrated first. The neurological score was assessed after 24 h of permanent stroke while using the Bederson Scale and 28-point neuro score. Animals that were subjected to p-MCAO (*n* = 7/group) exhibited severe neurological deficits (score 3 by Bederson scale) and a score of 11–15 (28-point composite score), which demonstrated an impaired cortical and striatal functions with associated diminished motor and sensory coordination (*p* < 0.001, Figure 3A,B). Treatment with A3 attenuated the neurobehavior deficits in a dose-dependent manner (*p* < 0.05, *p* < 0.01, Figure 3A,B). The 10 mg/kg dose exhibited a relatively better neuroprotective effect (*p* < 0.01, Figure 3A,B). Animals that were subjected to p-MCAO also revealed an increased water content (Figure 3C, *p* < 0.05, *n* = 5/group) compared to sham, depicting edema formation following ischemic stroke. A3 treatment mitigated the increased brain water content to a significant level (Figure 3C, *p* < 0.05). The focal cerebral ischemic model predominantly impairs the neocortex and striatal region, as previously demonstrated [38]. Therefore, TTC staining was performed in order to assess the cellular viability and neuronal damage (Figure 3D). As shown, A3 treatment reduced infarct area in a dose-dependent manner with 10 mg/kg showing maximum reduction of infarction (*p* < 0.001, One-way ANOVA followed by Bonferroni’s multiple comparisons test, *n* = 7/group) as compared to 5 mg/kg, with corrected infarct areas of 5.74% and 14.73%, respectively, as compared to 36.75% for p-MCAO (Figure 3D).

### 3.2. Effects of ATRA on A3 Mediated Neuroprotection

The activation of Nrf2 was inhibited by ATRA to demonstrate the possible role of Nrf2 in A3-mediated neuroprotection [16,17,18]. A dose of 5 mg/kg of ATRA resulted in aggravated oxidative stress markers as shown in Table 1. Further validation of these results was demonstrated through the neuro score (Figure 4A,B, *n* = 7/group) and corrected infarct area (Figure 4C, *n* = 7/group). ATRA abolished the protective effects of A3, causing intense neurobehavioral deficits, brain edema together with infarct area (Figure 4A–C, One-way ANOVA with Bonferroni’s multiple comparisons test, *n* = 7/group). These morphological changes were probed in the peri-infarct frontal cortex and striatum (Figure 4E), which was further investigated by Hematoxylin Eosin Staining (H & E). Atypical features were observed in the ipsilateral cortex and striatum, such as changes in neuronal shape and size (angular, swollen, and scalloped morphology), in staining (pyknosis/cytoplasmic eosinophilia, nuclear basophilia), and vacuolation (*p* < 0.001, Figure 4F, *n* = 5/group). These neurological damages were attenuated by A3 treatment as evidenced by an increase in the total number of intact neurons. Further to demonstrate glial cell activation after these infiltrations, we observed the immunoreactivity of Iba-1 as microglia cells (Figure 4G). Likely, higher Iba-1 positive cells were noticed in the frontal cortex of ischemic operated animals (*p* < 0.001). Moreover, ATRA exaggerated the deleterious effects of MCAO and diminished the neuroprotective effects of A3, thus impeding the Nrf2 and implicating its explicit involvement in the neuroprotective role of A3 (*p* < 0.05, *n* = 5/group).

### 3.3. Effect of A3 on Nrf2 Signaling Pathway

Furthermore, we antagonized the Nrf2 effects by ATRA using a dose of 5 mg/kg to investigate whether A3 effects are Nrf2-dependent. The neuroprotective effect of A3 was abolished by ATRA administration, as shown by ELISA analysis (Figure 5A). We performed immunohistochemistry and immunofluorescence, to further validate the transition of Nrf2 localization between cytoplasm and nucleus (Figure 5B,C). The results showed that A3 promoted the nuclear localization of Nrf2, as shown by DAPI/TRITC co-expression, whereas ATRA abrogated this transition, as demonstrated by immunohistochemical analysis. Furthermore, we also determined the protein expression of HO-1, which is downstream of the Nrf2 pathway, and a similar pattern of expression was observed for ATRA and A3 treated groups (Figure 5D,E). These results demonstrated that A3 could modulate both Nrf2 and HO-1 expression, while ATRA attenuated the effects of A3 on Nrf2.

### 3.4. A3 Attenuated Oxidative Stress Induced by MCAO

Table 1 summarizes the alterations in antioxidative enzymes following p-MCAO, A3, and ATRA treatment. p-MCAO induced ROS generation as opposed to sham, which is associated with nitric oxide (NO) amassing (13.35 ± 2.48 μmoles/mg) and the depletion of GST activity (13.51 ± 0.75 µmoles CDNB conjugate/min/mg of protein) and GSH level (14.28 ± 0.48 µmoles/mg of protein) in the cortex (*p* < 0.001). A3 post-treatment at different doses attenuated the downregulation of GSH (40 ± 1.23 µmoles/mg of protein) and GST (26.04 ± 2.11 µmoles CDNB conjugate/min/mg of protein) and reduced the level of NO (5.63 ± 0.92 μmoles/mg) when compared to the p-MCAO group.

### 3.5. Effect of A3 on LPO

The results of the TBARS test indicated a drastic rise in peroxides in MCAO, which could be restored by different doses of A3. The LPO concentration in the cortical homogenate was augmented to (118.8 ± 0.73 nM/min/mg protein) in comparison to the sham group (*p* < 0.001, Table 1). A3 at 10 mg/kg dose significantly (*p* < 0.001, Table 1) attenuated this rise (64.3 ± 0.52 nM/min/mg protein), an outcome that could be comparable to the sham group.

### 3.6. A3 Attenuated p-MCAO Induced Inflammatory Mediators in the Brain

In order to detect the possible involvement of TNF-α and JNK, ELISA analysis was performed and the results revealed the increased release of TNF-α in the p-MCAO group (*p* < 0.001) (Figure 6A), whereas A3 treatment significantly reduced the protein hyperexpression (*p* < 0.01, Figure 6A). Furthermore, the protein expression of TNF-α and p-JNK was validated by immunohistochemistry analysis (Figure 6B,C).

### 3.7. A3 Enhances the Anti-Inflammatory Capacity via the Nrf2 Signaling Pathway

A3′s antioxidative effect was further extrapolated to its anti-inflammatory role. We noticed comparatively elevated protein expression of TLR4 in the p-MCAO group as numerous reports have established the involvement of TLR4 in inflammation, which is eventually associated with the downstream inflammatory targets as p-NF-κB, and COX-2 [39,40], as shown in Figure 7A (*p* < 0.001). Additionally, *p*-NF-κB and COX-2 expression were also analyzed, and the results indicated that both are overly expressed in the p-MCAO group (*p* < 0.001, Figure 7B–D). A3 (10 mg/kg) mitigated both *p*-NF-κB and COX-2 expression along with TLR4.

## 4. Discussion

The present study was carried out in order to explore the neuroprotective effects of A3 in ischemic stroke-induced neurodegeneration. We demonstrated that A3 can mediate neuroprotective effects possibly by mitigating neuroinflammation, augmenting free-radical scavenging, and eventually reducing oxidative stress. We likely demonstrated aggravation of stroke due to wearing off the A3 mediated neuroprotective actions by employing an Nrf2 pathway inhibitor, ATRA. Moreover, A3 downregulated the expression of pro-inflammatory cytokines that are associated with enhanced endogenous anti-oxidants (Figure 8).

In humans, ischemic stroke occurs most commonly in part of the brain perfused by the middle cerebral artery MCA. Induced by intraluminal suture, focal cerebral ischemia is a well-established animal stroke model of clinical relevance, comprising of transient (t-MCAO) and permanent (p-MCAO) [41]. Previous studies have provided substantial research information to our current understanding regarding the pathophysiology of ischemic stroke [7,42]. Immediately after stroke onset, free radicals are released, which mobilizes the endogenous antioxidative machinery, of which, superoxide dismutase (SOD), glutathione s-transferase (GST), and reduced glutathione (GSH) are the first-line defense antioxidants [43]. Our results indicated that the post-ischemic A3 administration attenuated the LPO [44] and raised the depleted GST and GSH levels. GST and GSH have been extensively explored as having detoxifying effects against oxidative stress and they are pivotal to sustaining cellular homeostasis and scavenging free radicals [43,45]. Thus, sustaining the level of GST, GSH, along with low lipid peroxidation, might account for the antioxidant nature of A3.

As a part of endogenous antioxidant machinery, Nrf2 plays a pivotal role in redox homeostasis and regulates the transcription of antioxidant enzymes [18]. Following oxidative insult, including MCAO, Nrf2 translocates from the cytoplasm to the nucleus and activates transcription of several downstream antioxidant genes HO-1, NADPH quinone dehydrogenase-1 (NQO1), thioredoxin (TRX), GSH, SOD, and eventually protecting the cell from inflammation and apoptosis [46,47,48,49,50]. Several studies demonstrated the antioxidant and free radical scavenging effects of Nrf2 [51,52]. Conversely, Nrf2-deficient animals are remarkably more susceptible to brain insult and neurological damage than WT mice [53]. Our findings demonstrated the translocation of Nrf2 from the cytoplasm to the nucleus in the MCAO group in response to the oxidative stress signals, as the body’s natural defense mechanism. However, a further upregulation of Nrf2 levels was observed upon A3 treatment, associated with higher expression of downstream HO-1, demonstrating the antioxidant potential of A3. These effects, though, were subsided with Nrf2 inhibitor ATRA, indicates the possible role of Nrf2 in the A3 mediated neuroprotection. ATRA inhibits Nrf2 by several mechanisms: First, by augmenting the Nrf2-Keap1 dimer in the cytoplasm, which previously exaggerated intracerebral hemorrhage [17]. Second, ATRA could inhibit the binding of Nrf2 with ARE [16]. Third, the inhibitory effects of ATRA were attributed to its affinity for retinoic acid receptors (RAR-ά, -β, -γ) [16,18], thereby inhibiting the expression of ARE driven HO-1, and NQO1 activation. Based on these findings, we performed a series of experiments with different concentrations of ATRA (data not shown) and optimized a dose of 5 mg/kg, 30 min pre-surgery, which exerted maximal Nrf2 inhibition, thus aggravating outcomes of ischemic brain damage. Moreover, how A3 effects Nrf2 translocation needs further investigation.

A strong relationship between inflammation and oxidative stress has been revealed in both neuronal and non-neuronal models [54,55]. Furthermore, the activation of microglia, as indicated by Iba-1, further enhances the inflammatory process inducing neuronal death by numerous mechanisms [56]. Various studies reported that ligands of toll-like receptors (TLR4) trigger activation of certain stress-related kinases, including c-June N-terminal Kinases (JNK), which leads to the activation of the mitochondrial apoptotic pathway [57]. JNK (p-JNK) is involved not only in apoptotic mediated signaling, but also implicated in immune responses due to cross-talk with various proinflammatory cytokines [58]. Consistent with previous studies, we found an increased TLR4 and JNK expression, which was mitigated by A3 treatment, suggesting the anti-inflammatory role of A3.

Substantial evidence suggests the release of several pro-inflammatory cytokines, such as interleukin-6 (IL-6), interleukin-1β (IL-1β), and tumor necrotic factor-α (TNF-α) from glial cells [54,59]. Many transcription factors, including interferon regulatory factor-1 (IFN-1), NF-κB, and early growth response-1 factor (EGR1), also stimulate pro-inflammatory gene expression. which, in turn, leads to secondary neuronal apoptosis. Our study findings are similar to the previous reports, whereby inflammatory cytokines (TNF- α) are released from glial cells, starting a vicious cycle of activating other mediators of inflammation, such as *p*-NF-κB, COX-2, and iNOS during an early phase of ischemic insult, and that are attenuated by A3 administration.

## 5. Conclusions

In conclusion, our in-vivo findings demonstrate that A3 promotes neuronal survival by activating endogenous antioxidant Nrf2. Moreover, A3 also attenuated p-MCAO induced inflammatory cascade by the possible modulation of the Nrf2 pathway, which eventually accounts for its neuroprotective effects against neuronal inflammation. However, extensive exploration is still required in order to delineate the underlying protective mechanisms of A3.

## Figures and Tables

**Figure 1 biomolecules-10-00816-f001:**
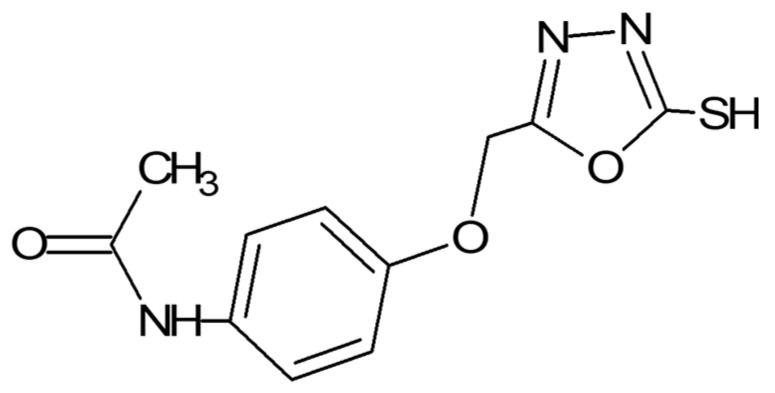
Structure of novel synthetic compound A3. *N*-{4-[(5-sulfanyl-1,3,4-oxadiazol-2-yl)methoxy]phenyl}acetamide.

**Figure 2 biomolecules-10-00816-f002:**
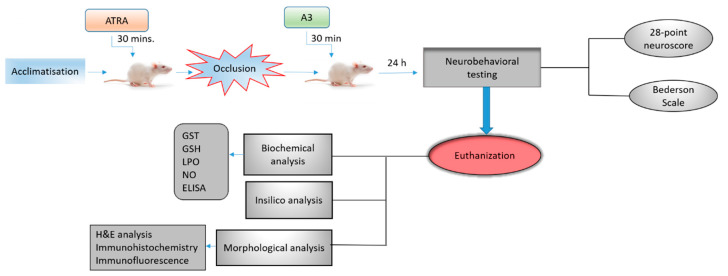
Schematic representation of the in vivo study design. Sham group: rats underwent surgery without filament insertion, vehicle-treated-p-MCAO group: rats treated with vehicle after p-MCAO induction, A3+p-MCAO (5 mg/kg) group: rats received 5 mg/kg of A3, 30 min after p-MCAO, A3+p-MCAO (10 mg/kg) group: rats received 10 mg/kg of A3, 30 min after p-MCAO, ATRA+p-MCAO (5 mg/kg) group: rats received 5 mg/kg of All trans-retinoic acid (ATRA) 30 min before p-MCAO, A3+ATRA+p-MCAO group: rats received 5 mg/kg of ATRA 30 min before p-MCAO and 10 mg/kg of A3 30 min after p-MCAO. Rats were sacrificed after 24 h of p-MCAO for further analysis.

**Figure 3 biomolecules-10-00816-f003:**
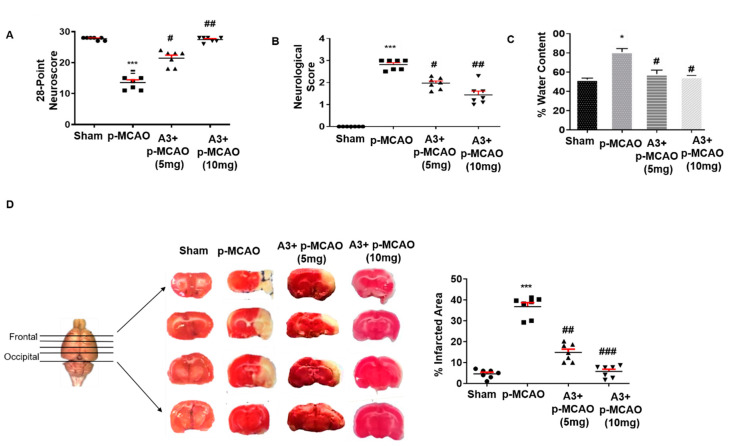
Effect of post-treatment dosage regimen of A3 on brain infarction and neuronal cell loss. (**A**) 28 points neuro scoring. *** *p* < 0.001 vs. sham, while ## *p* < 0.01 and # *p* < 0.05 vs. p-MCAO group (**B**) Bederson scale. A3+p-MCAO rats showed significantly reduced neurological deficits with a higher dose of A3 (## *p* < 0.01) as compared to p-MCAO rats (*n* = 7/group). *** *p* < 0.001 vs. sham, while ## *p* < 0.01 and # *p* < 0.05 vs p-MCAO group (**C**) Brain water content. p-MCAO showed significant % water content/edema formation, mitigated by A3 post-treatment (# *p* < 0.05, *n* = 5/group). * *p* < 0.05 vs. sham, while # *p* < 0.05 vs. p-MCAO group (**D**) % Infarct area was quantified by TTC staining started from the frontal cortex to the occipital cortex, which differentiates between ischemic and non-ischemic areas (*n* = 7/group). *** *p* < 0.001 vs. sham, while ## *p* < 0.01 vs. p-MCAO group.

**Figure 4 biomolecules-10-00816-f004:**
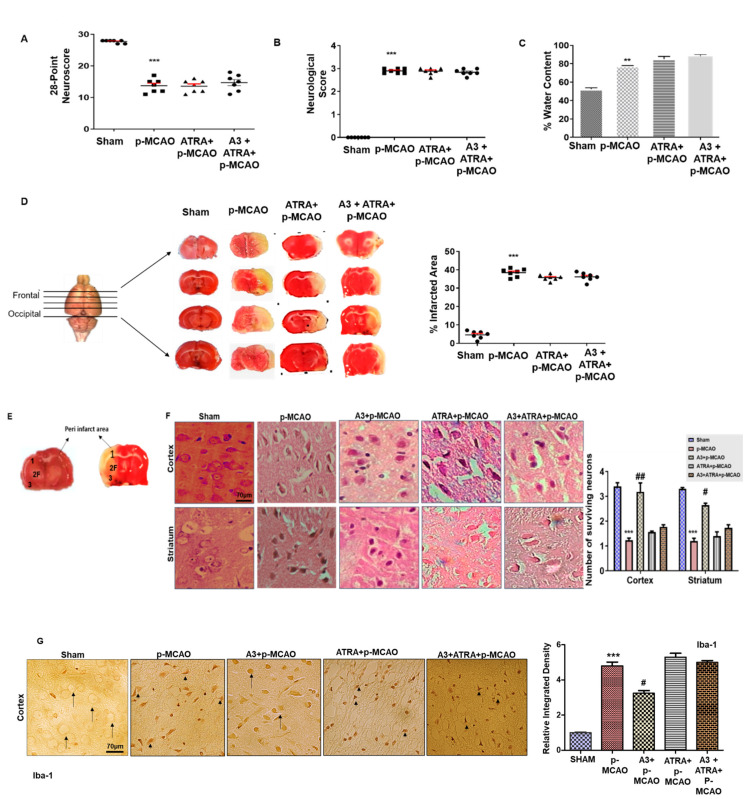
Effects of ATRA on A3 mediated neuroprotection. (**A**) 28 points composite scoring. (**B**) Bederson scale. ATRA+p-MCAO rats exhibited severe neurological impairment as with p-MCAO (*n* = 7/group). (**C**) The brain water content of the cerebral hemisphere in the ischemic side in each group (*n* = 5/group). ** *p* < 0.01 vs. sham group. ATRA+p-MCAO showed no significant changes in % water content/edema formation as compared to p-MCAO. (**D**) % Infarct area was quantified by TTC staining started from the frontal cortex to the occipital cortex, which differentiates between ischemic and non-ischemic areas (*n* = 7/group). (**E**) Coronal sections indicating frontal cortex (1), parietal cortex and insular cortex (2), and the piriform cortex (3). The analyzed region of interests indicated by 1 and F. (**F**) Representative images of hematoxylin and eosin staining in various groups. Scale bar = 70 μm. magnification 40×. (**G**) Representative images of Iba-1 immunostaining in the cortex. Scale bar = 70 μm, magnification 40×. The thin arrows indicated normal neuronal cells while the arrowhead indicates microglial phenotype. *** *p* < 0.001 vs. sham, # *p* < 0.05 and ## *p* < 0.01 vs. p-MCAO group.

**Figure 5 biomolecules-10-00816-f005:**
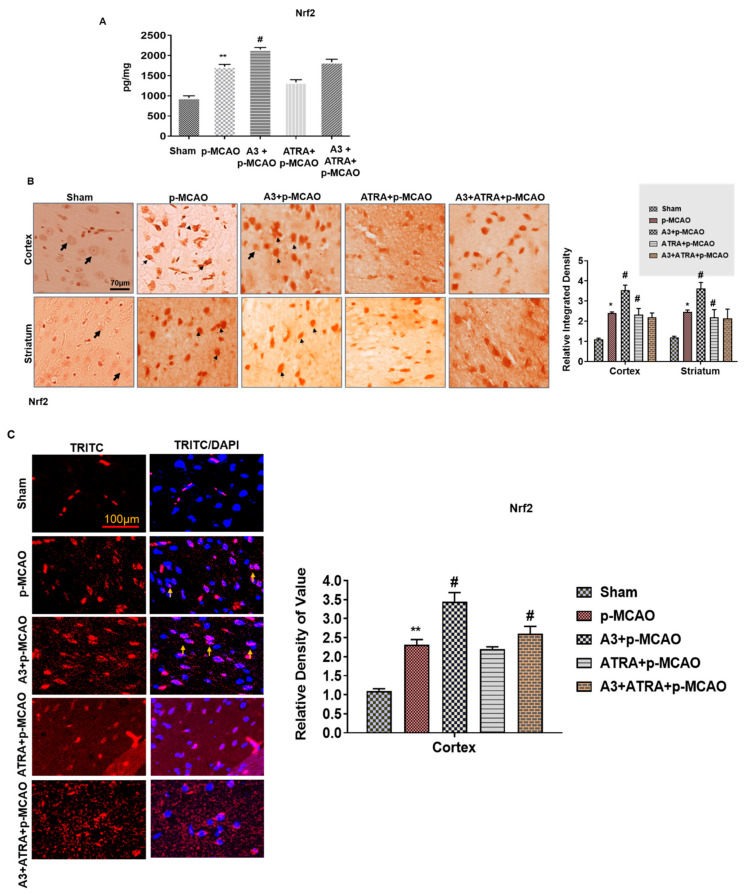
A3 modulates the antioxidant capacity of the brain via the Nrf2 signaling pathway (**A**) The protein expression of Nrf2 was quantified by ELISA in the brain cortex. ** *p* < 0.01 vs. sham while # *p* < 0.05 vs. p-MCAO (**B**) Immunohistochemical results for Nrf2 in cortex and striatum tissues of the brain. Bold arrows indicate a clear nucleus, while thin arrowhead shows nuclear staining, indicating Nrf2 translocation to the nucleus. Sham group shows the cytoplasmic localization of Nrf2 protein whereas A3 + p-MCAO treated group demonstrated nuclear localization. Scale bar = 70 μm, magnification 40×, (*n* = 5/group). * *p* < 0.05 vs. sham, # *p* < 0.05 vs. p-MCAO group. (**C**) Immunofluorescence reactivity of Nrf2 in cortex (*n* = 5/group). The aforementioned data is the representation of 3 numbers of experiments. Scale bar = 100 μm, magnification 40×. ** *p* < 0.01 vs. sham, # *p* < 0.05 vs. p-MCAO. Nrf2 was tagged by the TRITC signal, while the nucleus by DAPI. A3 increased the co-localization of DAPI and TRITC. (**D**) The protein expression of HO-1, as quantified by ELISA in the cortex. ** *p* < 0.01 vs. sham, ## *p* < 0.01 vs. p-MCAO (**E**) Immunohistochemistry results for HO-1 in cortex and striatum tissues of the brain. Scale bar = 50 μm, magnification 40×, (*n* = 5/group). *** *p* < 0.001 vs. sham, # *p* < 0.05 vs. p-MCAO group.

**Figure 6 biomolecules-10-00816-f006:**
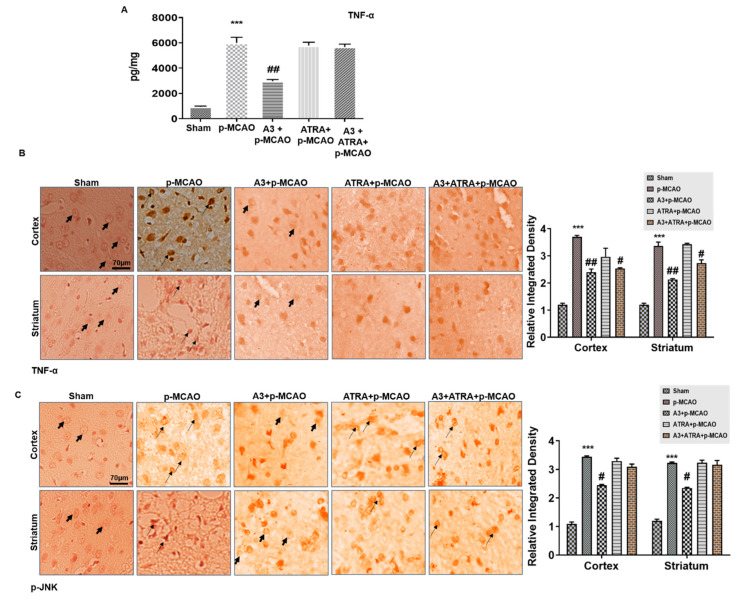
A3 ameliorated p-MCAO-induced neuroinflammation. (**A**) The release of TNF-α was quantified by ELISA in the cortex. *** *p* < 0.001 vs. sham, ## *p* < 0.01 vs. p-MCAO (*n* = 5/group). (**B**) Immunohistochemical results for TNF-α in cortex and striatum tissues of the brain. Scale bar = 70 μm, magnification 40× (*n* = 5/group). TNF-α exhibited cytoplasmic localization in both examined brain areas. (**C**) Immunohistochemistry results for p-JNK in cortex and striatum tissues of the brain. Scale bar = 70 μm, magnification 40×, (*n* = 5/group). p-JNK exhibited cytoplasmic localization. *** *p* < 0.001 vs. sham, # *p* < 0.05 or ## *p* < 0.01 vs. p-MCAO group. The bold arrow indicates no expression while thin arrow indicates cytoplasmic localization of TNF-α and p-JNK.

**Figure 7 biomolecules-10-00816-f007:**
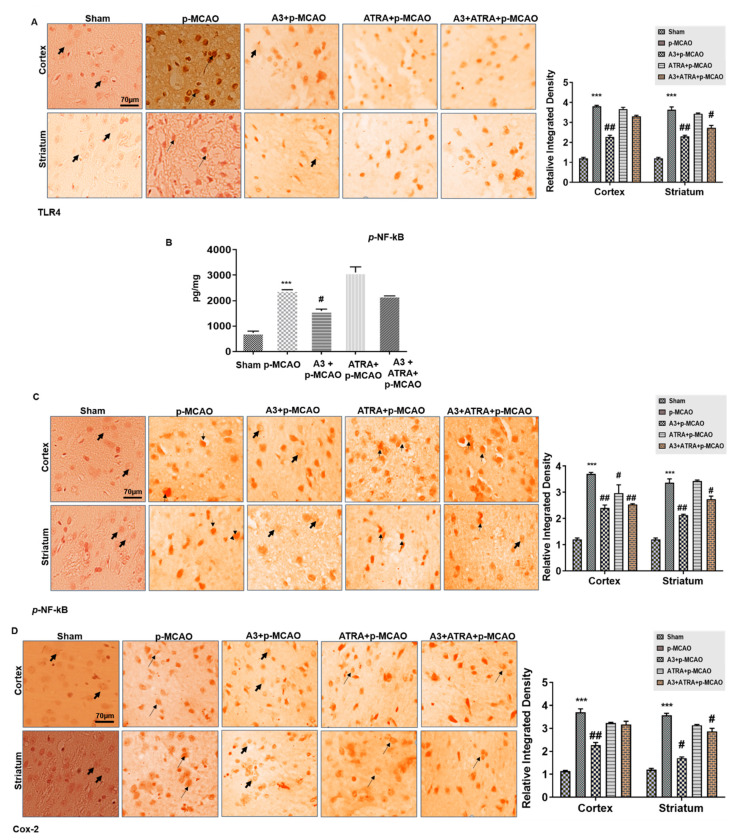
Effect of A3 on outcomes of p-MCAO induced inflammatory mediators. (**A**) Immunohistochemistry images of TLR4 in cortical and striatal tissues of the brain. Scale bar = 70 μm, magnification 40×, (*n* = 5/group). (**B**) Quantification of *p*-NF-κB expression by ELISA in the cortex. *** *p* < 0.001 vs. sham, # *p* < 0.05 vs. p-MCAO (**C**) Immunohistochemistry results for *p*-NF-κB in cortex and striatum tissues of the brain. Scale bar = 70 μm, magnification 40×, *(n* = 5/group). (**D**) Immunohistochemistry data for COX-2 in the cortex and striatum tissues of the brain. Scale Bar = 70 μm, magnification 40×, (*n* = 5/group). *p*-NF-κB exhibited nucleus localization while COX-2 and TLR4 manifested cytoplasmic localization in both tissues of the brain. *** *p* < 0.001 vs. sham, # *p* < 0.05 and ## *p* < 0.01 vs. p-MCAO group. The bold arrow indicates no expression while thin arrow indicates cytoplasmic localization of TLR4 and COX2. For p-NF-κB, the bold arrow indicates no nuclear expression while arrowhead indicates nuclear localization.

**Figure 8 biomolecules-10-00816-f008:**
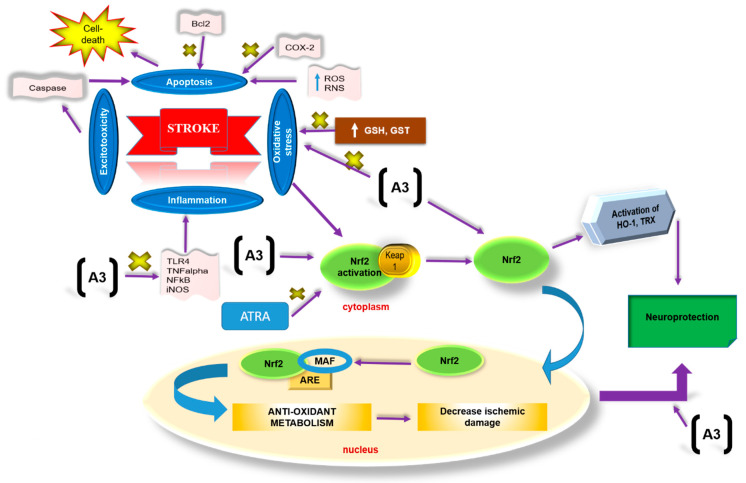
The graphical representation to elaborate on the possible mechanism underlying antioxidative and anti-inflammatory potential of A3 against the p-MCAO-induced brain injury. A3 causes Nrf2 activation and suppresses oxidative damage as indicated by elevated GST and GSH and reduced NO eventually leading to neuroprotection. ATRA blocks Nrf2 activation thereby leading to inhibition of endogenous antioxidative capacity. The majority of ROS and RNS are the products of oxygen metabolism in mitochondria. The superoxide anion produced is exported to the cytosol, and in the attempt to reduce the toxicity, some more reactive species are generated. These reactive species then cause degradation of DNA, lipids, and proteins. Under such conditions, the antioxidant response system (ARS) becomes activated to balance redox status. In our body, Nrf2 acts as a master regulator and is activated in response to stress via two mechanisms; a kinase-independent mechanism where reactive specie directly oxidize or nitrosylate the keap 1, a cysteine-rich protein that is bound to Nrf2 and thus releasing Nrf2. The other mechanism is kinase-dependent degradation, where a stressor such as glutamate, through activation of the Gq pathway, phosphorylate Nrf2. As a result, Nrf2 is transported into the nucleus where it displaces Bach1, a transcriptional repressor of ARE (antioxidant response elements), and heterodimerize with Maf and bind to ARE on DNA, eventually triggering transcriptional factors for NQO1, HO-1, and GST, thus increasing cellular defense against redox-modulators.

**Table 1 biomolecules-10-00816-t001:** Effect of A3 on oxidative enzymes.

	NO (μmoles/mg)	GST (µmoles CDNB Conjugate/min/mg of Protein)	GSH (µmoles/mg of Protein)	LPO (TBARS-nM/min/mg Protein)
**Sham**	4.089 ± 1.84	40.36 ± 1.09	83.4 ± 0.48	41.5 ± 0.42
**p-MCAO**	13.35 ± 2.48 **	13.51 ± 0.75 ***	14.28 ± 0.48 ***	118.8 ± 0.73 ***
**A3+p-MCAO (10mg/kg)**	5.63 ± 0.92 ##	26.04 ± 2.11 ###	40 ± 1.23 ###	64.3 ± 0.52 ###
**ATRA+p-MCAO**	17.84 ± 1.2	14.9 ± 1.83	20 ± 1.97	111.3 ± 0.52 ##
**A3+ATRA+p-MCAO**	15.45 ± 0.69	13.14 ± 2.06	18.4 ± 1.24	113 ± 0.42 ##

Symbols *** or ### indicates significant difference at *p* < 0.001, while ** and ## indicates significant difference at *p* < 0.01, respectively. *** vs. sham, # vs. p-MCAO group. The data are presented as mean ± SEM, with *n* = 5/group. The symbol * shows a significant difference vs. sham and # shows the significant difference vs. p-MCAO. Abbreviations: NO, nitric oxide; GST, glutathione S-transferase; GSH, reduced glutathione; LPO, lipid peroxidation; TBARS, thiobarbituric acid reactive substances.

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
