# Peer review of "Post-Treatment of Synthetic Polyphenolic 1,3,4 Oxadiazole Compound A3, Attenuated Ischemic Stroke-Induced Neuroinflammation and Neurodegeneration"

_biomolecules, 2020, doi:10.3390/biom10060816_

Round 1
Reviewer 1 Report
This is an interesting article but I have some questions and comments to be answered by the authors before giving my approval for publication,
1) In line 431, 432, the authors indicated: “ As shown by ELISA analysis (Figure 6A), the neuroprotective effect of A3 was abolished by ATRA administration.” This statement seems to be true. Noteworthy, the authors did not comment in the text that MCAO increases NFR-2 expression and A3 stimulates this expression above the levels found after MCAO. This is not surprising since the role of reactive oxygen species in the modulation of Nrf2 following ischemic reperfusion Injury has been described. The authors should give an explanation for these results since they suggest that there might be another pathway responsible for the observed results. In other words, which is the difference of having an increase of NRF-2 expression of 80pg/mL of 110pg/mL in respect to sham 45pg/mL, in terms of control of the expression of important enzymes protecting in this model (Fig 6 A)?, for example, HO-1 which major changes are found between MCAO and Sham samples. Please discuss it.
2) NO· production units should be corrected to be shown as μmoles/mg of tissue instead of μmoles/L.
3) NFR-2, HO-1, TNF-2, TLR-4 levels, as measured in pg/mL, are meaningless since the measurements are not made in circulation, cell culture media or liquid samples. They are measured from tissue/mass and therefore normalized by mg of protein.
4) The authors studied the expression levels of TLR4 and Cox-2 and they later used bioinformatics tools to study the possible effect on the catalytic activity of COX-2, HO-1, p-JNK, p-NF-κB, and Nrf2. This is a very weak approximation since in the absence of kinetic or binding data supporting these interactions, these results are not be validated approximations that only suggest a possible interaction but they are not proved. Docking data should not be used in the absence of data that supports the docking data. I suggest the authors eliminate all the information regarding the bioinformatic analysis or to include kinetic or binding data supporting the proposed models and the interactions.
5) The authors should highlight in the text other possible pathways upstream activation of NRF-2, as those in charge of RNS and ROS production that might be blocked by A3 and some of them commented in the discussion. If it is possible for authors, they should measure some of them to complete the manuscript. i.e. nitric oxide synthase activity.
Minor comments:
- Although it is supposed, slices shown in figure 3D and 4D represent different coronal slices of the brain. The authors should indicate the area of the brain that are represented, and a legend should be included. Please, Increase the resolution of figure 3D and 4D, 5B, and 5D and expand these panels.
- Figures: 3A, 3C, 4C, 4E, 4F, 5B, 6B, 6E, 7B are out of the margins of the manuscript. Please reorganize them to fit inside the text of the manuscript.
Author Response
Reviewer 1
This is an interesting article but I have some questions and comments to be answered by the authors before giving my approval for publication,
Q1: In line 431, 432, the authors indicated: “ As shown by ELISA analysis (Figure 6A), the neuroprotective effect of A3 was abolished by ATRA administration.” This statement seems to be true. Noteworthy, the authors did not comment in the text that MCAO increases NFR-2 expression and A3 stimulates this expression above the levels found after MCAO. This is not surprising since the role of reactive oxygen species in the modulation of Nrf2 following ischemic reperfusion Injury has been described. The authors should give an explanation for these results since they suggest that there might be another pathway responsible for the observed results. In other words, which is the difference of having an increase of NRF-2 expression of 80pg/mL of 110pg/mL in respect to sham 45pg/mL, in terms of control of the expression of important enzymes protecting in this model (Fig 6 A)?, for example, HO-1 which major changes are found between MCAO and Sham samples. Please discuss it.
Answer: We are thankful to the respected reviewer for his deep and thorough reviewing of our manuscript. We appreciate the sound background knowledge and expertise of the respected reviewer.
As a part of endogenous antioxidant machinery, Nrf2 plays a pivotal role in redox homeostasis and regulates the transcription of antioxidant enzymes [1]. Following oxidative insult including MCAO, Nrf2 translocates from the cytoplasm to the nucleus and activates transcription of several downstream antioxidant genes HO-1, NADPH quinone dehydrogenase-1 (NQO1), thioredoxin (TRX), GSH, SOD and eventually protecting the cell from inflammation and apoptosis [2-6]. Several studies demonstrated the antioxidant and free radical scavenging effects of Nrf2 [7,8]. Conversely, Nrf2-deficient animals are remarkably more susceptible to brain insult and neurological damage than WT mice [9]. Our findings demonstrated the translocation of Nrf2 from the cytoplasm to the nucleus in MCAO group in response to the oxidative stress signals, as body’s natural defense mechanism, however, upon A3 treatment, a further upregulation of Nrf2 levels was observed, associated with that of phase II detoxifying enzyme HO-1, demonstrating the antioxidant potential of A3. These effects, however, were subsided with Nrf2 inhibitor ATRA, indicating the possible role of Nrf2 in the A3 mediated neuroprotection. ATRA inhibits Nrf2 by several mechanisms: First, by augmenting the Nrf2-Keap1 dimer in the cytoplasm, which previously exaggerated intracerebral hemorrhage [10]. Second, ATRA could inhibit the binding of Nrf2 with ARE [11]. Third, the inhibitory effects of ATRA were attributed to its affinity for retinoic acid receptors (RAR-ά, -β, -γ) [11,1], thereby inhibiting the expression of ARE driven HO-1, and NQO1 activation. Based on these findings, we performed a series of experiments with different concentrations of ATRA (data not shown) and optimized a dose of 5 mg/kg, 30 min pre-surgery, which exerted maximal Nrf2 inhibition, thus aggravating outcomes of ischemic brain damage. Moreover, how A3 effects on Nrf2 translocation needs further investigation.
We have added this discussion (line 560-580) in the revised manuscript and highlighted the changes we made.
References:
- Liu, D.; Xue, J.; Liu, Y.; Gu, H.; Wei, X.; Ma, W.; Luo, W.; Ma, L.; Jia, S.; Dong, N., et al. Inhibition of NRF2 signaling and increased reactive oxygen species during embryogenesis in a rat model of retinoic acid-induced neural tube defects. NeuroToxicology 2018, 69, 84-92.
- Luo, J.-F.; Shen, X.-Y.; Lio, C.K.; Dai, Y.; Cheng, C.-S.; Liu, J.-X.; Yao, Y.-D.; Yu, Y.; Xie, Y.; Luo, P., et al. Activation of Nrf2/HO-1 Pathway by Nardochinoid C Inhibits Inflammation and Oxidative Stress in Lipopolysaccharide-Stimulated Macrophages. Frontiers in Pharmacology 2018, 9, 911-911.
- Suzuki, T.; Yamamoto, M. Molecular basis of the Keap1–Nrf2 system. Free Radical Biology and Medicine 2015, 88, 93-100.
- Pickering, A.M.; Linder, R.A.; Zhang, H.; Forman, H.J.; Davies, K.J.A. Nrf2-dependent induction of proteasome and Pa28αβ regulator are required for adaptation to oxidative stress. The Journal of biological chemistry 2012, 287, 10021-10031.
- Walters, D.M.; Cho, H.-Y.; Kleeberger, S.R. Oxidative Stress and Antioxidants in the Pathogenesis of Pulmonary Fibrosis: A Potential Role for Nrf2. Antioxidants & Redox Signaling 2007, 10, 321-332.
- Habtemariam, S. The Nrf2/HO-1 Axis as Targets for Flavanones: Neuroprotection by Pinocembrin, Naringenin, and Eriodictyol. Oxidative Medicine and Cellular Longevity 2019, 2019.
- Reisman, S.A.; Csanaky, I.L.; Aleksunes, L.M.; Klaassen, C.D. Altered Disposition of Acetaminophen in Nrf2-null and Keap1-knockdown Mice. Toxicological Sciences 2009, 109, 31-40.
Q2: NO production units should be corrected to be shown as μmoles/mg of tissue instead of μmoles/L.
Answer: Thank you very much for the correction. We have corrected this in the revised manuscript, thank you very much.
Q3: NFR-2, HO-1, TNF-2, TLR-4 levels, as measured in pg/mL, are meaningless since the measurements are not made in circulation, cell culture media or liquid samples. They are measured from tissue/mass and therefore normalized by mg of protein.
Answer: We are thankful to the respected reviewer for the constructive comments by raising some critical concerns, which will make our manuscript more valuable. Typically, the ELISA kit is provided with known concentrations of standard samples in pg/mL, ng/mL or µg/mL. ELISA data is then graphed with optical density vs these known concentrations of target proteins (standards provided by the manufacturer) to obtain a calibration curve (standard curve). The unknown concentrations of target proteins in our tissue lysates are then calculated by comparing with this standard curve thereby making our results accurate and reproducible. As concentrations are calculated by comparing with that of standards provided (by the manufacturer), which were in pg/mL in our case, therefore final concentration calculated are in the same units, which is picogram of protein/mL of tissue lysate.
Q4: The authors studied the expression levels of TLR4 and Cox-2 and they later used bioinformatics tools to study the possible effect on the catalytic activity of COX-2, HO-1, p-JNK, p-NF-κB, and Nrf2. This is a very weak approximation since in the absence of kinetic or binding data supporting these interactions, these results are not be validated approximations that only suggest a possible interaction but they are not proved. Docking data should not be used in the absence of data that supports the docking data. I suggest the authors eliminate all the information regarding the bioinformatic analysis or to include kinetic or binding data supporting the proposed models and the interactions.
Answer: We agree with the worthy comments of the reviewer to raise this valuable and scientific point. The binding energies along with the number of hydrogen bonds present a good indication of possible interactions among the proteins and ligands and this was further interlinked with in-vivo experimentation similar to previously reported data [1,2]. Fig 8, as well as Table 2, clearly depicts the amino acids forming bonds with the ligands and the number of hydrogen bonds and binding energies indicates the strength of the bond formation.
References:
- Imran, M.; Al Kury, L.T.; Nadeem, H.; Shah, F.A.; Abbas, M.; Naz, S.; Khan, A.-u.; Li, S. Benzimidazole Containing Acetamide Derivatives Attenuate Neuroinflammation and Oxidative Stress in Ethanol-Induced Neurodegeneration. Biomolecules 2020, 10, 108.
- Faheem, M.; Khan, A.-U.; Nadeem, H.; Ali, F. Computational and Phamracological Evaluation Of Heterocyclic 1,3,4-Oxadiazole and Pyrazoles Novel Derivatives For Toxicity Assesment, Tumor Inhibition, Antioxidant, Analgesic And Anti-inflammatory Actions. FARMACIA 2018, 66, 909-919.
Q5: The authors should highlight in the text other possible pathways upstream activation of NRF-2, as those in charge of RNS and ROS production that might be blocked by A3 and some of them commented in the discussion. If it is possible for authors, they should measure some of them to complete the manuscript. i.e. nitric oxide synthase activity.
The majority of ROS and RNS are the products of oxygen metabolism in mitochondria. The superoxide anion produced is exported to the cytosol, and in the attempt to reduce the toxicity, some more reactive species are generated. These reactive species then cause degradation of DNA, lipids, and proteins. Under such conditions, the antioxidant response system (ARS) becomes activated to balance redox status. In our body, Nrf2 acts as a master regulator and is activated in response to stress via two mechanisms; a kinase-independent mechanism where reactive species directly oxidize or nitrosylate the keap 1, a cysteine-rich protein that is bound to Nrf2 and thus releasing Nrf2. The other mechanism is kinase-dependent degradation where a stressor such as glutamate, through activation of the Gq pathway, phosphorylate Nrf2. As a result, Nrf2 is transported into the nucleus where it displaces Bach1, a transcriptional repressor of ARE (antioxidant response elements), and heterodimerize with Maf and bind to ARE on DNA, eventually triggering transcriptional factors for NQO1, HO-1, and GST, thus increasing cellular defense against redox-modulators [1,2,3].
We have added this (line (526-538) in the revised manuscript and highlighted the changes we made.
- Fraunberger, E.; Scola, G.; Laliberté, V.; Duong, A.; Andreazza, A. Redox Modulations, Antioxidants, and Neuropsychiatric Disorders. Oxidative Medicine and Cellular Longevity 2016, 2016, 1-14.
- Ray, P. D., Huang, B. W., & Tsuji, Y. (2012). Reactive oxygen species (ROS) homeostasis and redox regulation in cellular signaling. Cellular signalling, 24(5), 981–990. https://doi.org/10.1016/j.cellsig.2012.01.008
- Vargas-Mendoza, N., Morales-González, Á., Madrigal-Santillán, E. O., Madrigal-Bujaidar, E., Álvarez-González, I., García-Melo, L. F., ... & Morales-Gonzalez, J. A. (2019). Antioxidant and adaptative response mediated by Nrf2 during physical exercise. Antioxidants, 8(6), 196.
Minor comments:
- Although it is supposed, slices shown in figure 3D and 4D represent different coronal slices of the brain. The authors should indicate the area of the brain that are represented, and a legend should be included. Please, Increase the resolution of figure 3D and 4D, 5B, and 5D and expand these panels ( I will do this).
Answer: Thank you very much for raising the point. We have inserted the brain matrix diagram and also included the legend. We have also increased the resolution of Figure 3D, 4D, 5B and 5D. Please find in the revised manuscript.
- Figures: 3A, 3C, 4C, 4E, 4F, 5B, 6B, 6E, 7B are out of the margins of the manuscript. Please reorganize them to fit inside the text of the manuscript (I will do this).
Answer: Again we are thankful to the respected reviewer. We have reorganized Figures: 3A, 3C, 4C, 4E, 4F, 5B, 6B, 6E, 7B. Please find in the revised manuscript.
Reviewer 2 Report
The article is in good form and may be accepted after minor correction
style of references
add new references 2019-2020
check the typos
check minor correction in language
Author Response
The article is in good form and may be accepted after minor correction
style of references
add new references 2019-2020
check the typos
check minor correction in language
Answer: We are thankful to the respected reviewer for his deep and thorough reviewing of our manuscript. We appreciate the sound background knowledge and expertise of the respected reviewer. We have rechecked our manuscript for all the issues you have mentioned above and corrected in the revised manuscript. We have tried to adopt the uniform style of references and a standard format of journal requirement, added new references 2019-2020, tried to overcome the typos, and critically checked for minor language correction. Thank you very much
Reviewer 3 Report
Presented paper is designed to prove post-Treatment of Synthetic Polyphenolic 1,3,4 2 Oxadiazole Compound A3, Attenuated Ischemic Stroke-Induced Neuroinflammation and Neurodegeneration. Till now there is no real treatment of ischemic stroke, so study is interesting, however several questions and notes should be clarified or explained.
Design of the study is based mainly on the protective effect of A3, compound with the prospective antioxidant and antiinflammatory properties. Reviewer misses real pharmacodynamic data on this compound, its proved bioprotective properties , see citation 18, -or the data are not complete and citation source does not seems as reputated and internationally recognized. In general, citation of references manifests several errors, some of them are obsolete, not very recent.
Referee misses the data from control, not sham operated animals, since surgery itself is stressfull and changes inflammatory response.
What was percentage of the animals within the exclusion criteria.
Experiments with the aim to check oxidative stress lack the complexity of the biological processes that changes redox state of the cells(tissue). Detection of GHS and GST cover only restricted part of oxidative stress, determination of lipoperoxidation by MDA detection in very unspecific.
On the other hand, reviewer appreciates results which cover histological damage
Author Response
Reviewer 3:
Presented paper is designed to prove post-Treatment of Synthetic Polyphenolic 1,3,4 2 Oxadiazole Compound A3, Attenuated Ischemic Stroke-Induced Neuroinflammation and Neurodegeneration. Till now there is no real treatment of ischemic stroke, so study is interesting, however several questions and notes should be clarified or explained.
Q1: Design of the study is based mainly on the protective effect of A3, compound with the prospective antioxidant and antiinflammatory properties. Reviewer misses real pharmacodynamic data on this compound, its proved bioprotective properties , see citation 18, -or the data are not complete and citation source does not seems as reputated and internationally recognized. In general, citation of references manifests several errors, some of them are obsolete, not very recent.
Answer: Thank you very much for your worthy remarks and corrections. Yes, we agree with the respected reviewer valuable point. The previous study demonstrated the cytotoxic effect of these derivatives in various cancer lines including the brain [1]. Moreover, previously our lab members showed lethality assay and acute toxicity evaluation, whereas no mortality was recorded up to the dose of 750 mg/kg [2]. Based upon we proceeded to animal studies. Moreover, we implied low doses of this derivative. We added this description in the introduction part (Line 94-104) and highlighted the changes. The citation of references has also been updated. Please find in the revised manuscript.
References:
- Ahsan, M.J.; Choupra, A.; Sharma, R.K.; Jadav, S.S.; Padmaja, P.; Hassan, M.; Al-Tamimi, A.; Geesi, M.H.; Bakht, M.A. Rationale design, synthesis, cytotoxicity evaluation, and molecular docking studies of 1, 3, 4-oxadiazole analogues. Anti-Cancer Agents in Medicinal Chemistry (Formerly Current Medicinal Chemistry-Anti-Cancer Agents) 2018, 18, 121-138.
- Faheem, M.; Khan, A.-U.; Nadeem, H.; Ali, F. Computational and Phamracological Evaluation Of Heterocyclic 1,3,4-Oxadiazole and Pyrazoles Novel Derivatives For Toxicity Assesment, Tumor Inhibition, Antioxidant, Analgesic And Anti-inflammatory Actions. FARMACIA 2018, 66, 909-919.
Q2; Referee misses the data from control, not sham operated animals, since surgery itself is stressfull and changes inflammatory response.
Answer: Thank you very much for the reading and reviewing our manuscript which will help us to improve it to a better scientific level. Respected reviewer, measuring response against sham-operated animals is the standard approach as reported in various similar research designs [1,2], and in fact, many research articles designated sham as sham control [3]. We agree with the respected reviewer, the surgery changes the inflammatory response, and therefore the sham-operated animal rule out any harm which surgery itself can evoke without suture insertion.
- Chen, L.; Huang, K.; Wang, R.; Jiang, Q.; Wu, Z.; Liang, W.; Guo, R.; Wang, L. Neuroprotective Effects of Cerebral Ischemic Preconditioning in a Rat Middle Cerebral Artery Occlusion Model: The Role of the Notch Signaling Pathway. Biomed Res Int 2018, 2018, 8168720-8168720.
- Yu, J.; Wang, W.-n.; Matei, N.; Li, X.; Pang, J.-w.; Mo, J.; Chen, S.-p.; Tang, J.-p.; Yan, M.; Zhang, J.H. Ezetimibe Attenuates Oxidative Stress and Neuroinflammation via the AMPK/Nrf2/TXNIP Pathway after MCAO in Rats. Oxidative Medicine and Cellular Longevity 2020, 2020.
- Li, W.; Huang, R.; Shetty, R.A.; Thangthaeng, N.; Liu, R.; Chen, Z.; Sumien, N.; Rutledge, M.; Dillon, G.H.; Yuan, F., et al. Transient focal cerebral ischemia induces long-term cognitive function deficit in an experimental ischemic stroke model. Neurobiology of Disease 2013, 59, 18-25.
Q3: What was percentage of the animals within the exclusion criteria.
Answer: Nearly 15% of animals fell in the exclusion criteria, as we performed nearly 115 surgeries (excluding dead ones), of which 97 were included in the study.
Q4: Experiments with the aim to check oxidative stress lack the complexity of the biological processes that changes redox state of the cells(tissue). Detection of GHS and GST cover only restricted part of oxidative stress, determination of lipoperoxidation by MDA detection in very unspecific.
Answer: We agree with the respected reviewer viewpoint. We aimed at measuring initially the first-line defense system of the body which comprised mainly of GST, GSH, and LPO, owing to the availability of our resources. Also, a major antioxidant defense mechanism, Nrf2 along with HO-1 was studied deeply which is an advanced antioxidant stress determinant and quite efficiently portrays the oxidative stress state level of the body. Nrf2 is a key transcription factor that regulates cells in response to invaders and oxidative damage. Nrf2 is the most important activator of AREs. Under oxidative stress conditions, Nrf2 dissociates from Keap1, translocates into the nucleus, combines with the Maf protein to form a heterodimer, and recognizes the appropriate ARE sequence activating SOD, CAT, GST, HO-1, and NQO1. Nrf2 is a key transcription factor that regulates cells in response to invaders and oxidative damage. Degradation and inhibition of Nrf2 causes cells to become more sensitive, which then leaves them vulnerable to damage, even in low-stress environments. [3,4].
- 1. Ma, Q. Role of Nrf2 in Oxidative Stress and Toxicity. Annual Review of Pharmacology and Toxicology 2013, 53, 401-426.
- Chen, B.; Lu, Y.; Chen, Y.; Cheng, J. The role of Nrf2 in oxidative stress-induced endothelial injuries. J Endocrinol 2015, 225, R83-R99.
On the other hand, reviewer appreciates results which cover histological damage
Thank you very much for your kind words.
Reviewer 4 Report
In the manuscript “Post-treatment of synthetic polyphenolic 1,3,4 oxadiazole compound A3, attenuated ischemic stroke-induced neuroinflammation and neurodegeneration” the authors reported the neuroprotective effects of the synthetic polyphenolic drug A3 in a model of ischemic stroke, the permanent middle cerebral artery occlusion (MCAO). They investigated the anti-oxidant and anti-inflammatory effects of this drug and the role of Nrf2 pathway activation in A3 mediated neuroprotection.
Major Revision
The manuscript shows some writing mistakes. Please check spelling, punctuation, typographical emphasis, abbreviations through all the paper, in order to be consistent (i.e ELISA instead of Elisa or abbreviations reported in more than one paragraph). There are also some part of the text with different colors or underlined, please check and uniform it.
Introduction
the description of the known modifications and effects of Nrf2 and ATRA in ischemia needs to be improved in the first session of the manuscript.
Methods
- The description of the method for immunofluorescence analysis is poor. Please add more details, i.e. the dilution of antibodies, the type of secondary antibody, the precise protocol etc.
- The details for immunohistochemistry analysis quantification need to be augmented.
Results and figures.
- The description of the results should to be more precise and detailed, i.e. the comparison between sham and MCAO, and the effects of treatment (A3 or ATRA) need to be underlined in all the paragraphs
- the captions of the figures need to be simplified i.e. some details of the methods are already described in the appropriate section
- the quality of the photographs of immunohistochemistry are poor. Please provide higher quality images.
- The paragraph 3.6 should be moved to 3.3, in order to emphasize the effect of Nrf2 inhibition on A3 effects.
- The table and figures should appear immediately after the first citation, please check
- The cellular localization (nuclear or cytoplasmic) of the immunohistochemistry signals cannot be clearly determined. Also the difference between neurons in hematoxylin /eosin staining cannot be appreciated. Please provide some additional data (i.e. new immunofluorescence studies or double staining) to confirm these statements.
Discussion
- The effect of ATRA on neuroprotection is referred as “exaggerated the deleterious effects of MCAO (line362)” and “aggravating outcomes of ischemic brain damage” (line556). Neurological scores and infarcted area, as well as oxidative stress markers, of MCAO +ATRA animals are not different if compared to MCAO group. Could you explain the basis of the reported statements?
- In figure 6 Nrf2 and HO-1 appear elevated already in MCAO group compared to sham, and the treatment with A3 just increased these values. Which is the role of Nfr2 increase, beneficial or detrimental? Please comment and justify these data in the discussion.
- in several parameters, such as Nrf2 immunofluorescence or TLR4 in the stratum or p-NF-kB, the group of A3+MCAO+ATRA groups appear significant different from MCAO group, such as A3+MCAO animals. Could you explain these discrepancies? Please add some information in the discussion.
Author Response
Reviewer 4:
In the manuscript “Post-treatment of synthetic polyphenolic 1,3,4 oxadiazole compound A3, attenuated ischemic stroke-induced neuroinflammation and neurodegeneration” the authors reported the neuroprotective effects of the synthetic polyphenolic drug A3 in a model of ischemic stroke, the permanent middle cerebral artery occlusion (MCAO). They investigated the anti-oxidant and anti-inflammatory effects of this drug and the role of Nrf2 pathway activation in A3 mediated neuroprotection.
Major Revision
Q: The manuscript shows some writing mistakes. Please check spelling, punctuation, typographical emphasis, abbreviations through all the paper, in order to be consistent (i.e ELISA instead of Elisa or abbreviations reported in more than one paragraph). There are also some part of the text with different colors or underlined, please check and uniform it.
We are thankful to the respected reviewer for his deep and thorough reviewing of our manuscript. We appreciate the sound background knowledge and expertise of the respected reviewer. We have tried our level best to cover all the spelling mistakes and typos and to observe uniformity.
Introduction
the description of the known modifications and effects of Nrf2 and ATRA in ischemia needs to be improved in the first session of the manuscript.
Answer: Thank you very much for raising the concern. Additional information and changes have been made (Line 77-90) as follow:
In the nucleus, Nrf2 binds to antioxidant response elements (AREs) and triggers transcription machinery of endogenous protective enzymes including antioxidant genes, phase II detoxifying enzyme, anti-inflammatory genes, and molecular chaperones, and further induces several cellular protective functions in numerous inflammatory reactions, malignant tumors, cardiovascular diseases and respiratory diseases [1]. Hence, owing to its imperative role in cellular resistance to exogenous toxic substances and oxidative stress, it may act as a prominent therapeutic target in acute ischemic stroke.
All trans-retinoic acid (ATRA) is a vitamin-A derivative and binds to retinoic acid receptors (RARs) and demonstrates immunoregulatory activities in numerous models [2]. Conflicting activities have been attributed to ATRA, including some neuroprotective effects [3,4]. However, researchers have also documented the inhibitory effects of ATRA on the Nrf2-ARE pathway where it either augmented Nrf2-Keap dimer in the cytoplasm or inhibited activation of ARE [5,6]. In another study, the inhibitory effects of ATRA were reported owing to retinoic acid receptors (RAR) ultimately inhibiting ARE driven genes [7].
The changes made were highlighted in the revised manuscript.
References:
- Kensler, T.W.; Wakabayashi, N.; Biswal, S. Cell Survival Responses to Environmental Stresses Via the Keap1-Nrf2-ARE Pathway. Annual review of pharmacology and toxicology 2007, 47, 89-116.
- Lu, L.; Lan, Q.; Li, Z.; Zhou, X.; Gu, J.; Li, Q.; Wang, J.; Chen, M.; Liu, Y.; Shen, Y. Critical role of all-trans retinoic acid in stabilizing human natural regulatory T cells under inflammatory conditions. Proceedings of the National Academy of Sciences 2014, 111, E3432-E3440.
- Li, M.; Tian, X.; An, R.; Yang, M.; Zhang, Q.; Xiang, F.; Liu, H.; Wang, Y.; Xu, L.; Dong, Z. All-Trans Retinoic Acid Ameliorates the Early Experimental Cerebral Ischemia–Reperfusion Injury in Rats by Inhibiting the Loss of the Blood–Brain Barrier via the JNK/P38MAPK Signaling Pathway. Neurochemical research 2018, 43, 1283-1296.
- Cai, W.; Wang, J.; Hu, M.; Chen, X.; Lu, Z.; Bellanti, J.A.; Zheng, S.G. All trans-retinoic acid protects against acute ischemic stroke by modulating neutrophil functions through STAT1 signaling. Journal of Neuroinflammation 2019, 16, 175.
- Wang, X.J.; Hayes, J.D.; Henderson, C.J.; Wolf, C.R. Identification of retinoic acid as an inhibitor of transcription factor Nrf2 through activation of retinoic acid receptor alpha. Proceedings of the National Academy of Sciences 2007, 104, 19589-19594.
- Yin, X.-P.; Zhou, J.; Wu, D.; Chen, Z.-Y.; Bao, B. Effects of that ATRA inhibits Nrf2-ARE pathway on glial cells activation after intracerebral hemorrhage. International journal of clinical and experimental pathology 2015, 8, 10436-10443.
- Liu, D.; Xue, J.; Liu, Y.; Gu, H.; Wei, X.; Ma, W.; Luo, W.; Ma, L.; Jia, S.; Dong, N., et al. Inhibition of NRF2 signaling and increased reactive oxygen species during embryogenesis in a rat model of retinoic acid-induced neural tube defects. NeuroToxicology 2018, 69, 84-92.
Methods
- The description of the method for immunofluorescence analysis is poor. Please add more details, i.e. the dilution of antibodies, the type of secondary antibody, the precise protocol etc. The details for immunohistochemistry analysis quantification need to be augmented.
Answer: Again we are thankful to the respected reviewer. The detailed protocol of immunofluorescence has been added. Line (293-308). Furthermore, quantification has been elaborated further (286-291). The changes made were highlighted in the revised manuscript.
Results and figures.
- The description of the results should to be more precise and detailed, i.e. the comparison between sham and MCAO, and the effects of treatment (A3 or ATRA) need to be underlined in all the paragraphs.
Answer: Thank you very much. We have underlined some of the descriptions of the results for the respected reviewer. We are endorsing the respected reviewer viewpoint.
- the captions of the figures need to be simplified i.e. some details of the methods are already described in the appropriate section
Answer: Respected reviewer, the captions have been simplified as directed.
the quality of the photographs of immunohistochemistry are poor. Please provide higher quality images.-
Answer: The Quality of Immuno figures has been changed or modified. Please find in the revised manuscript.
- The paragraph 3.6 should be moved to 3.3, in order to emphasize the effect of Nrf2 inhibition on A3 effects.
Answer: Paragraph 3.6 has been moved to 3.3. Thank you very much
- The table and figures should appear immediately after the first citation, please check
Answer: Respected reviewer, tables, and figures have been cross-checked.
- The cellular localization (nuclear or cytoplasmic) of the immunohistochemistry signals cannot be clearly determined. Also the difference between neurons in hematoxylin /eosin staining cannot be appreciated. Please provide some additional data (i.e. new immunofluorescence studies or double staining) to confirm these statements.
Answer: We have added new immunohistochemistry figures, and have also modified the hematoxylin /eosin staining. We have added arrow directions to both cytoplasm and nucleus. Please find in the revised manuscript. All changes made are highlighted (please refer to figure legends)
Discussion
- The effect of ATRA on neuroprotection is referred as “exaggerated the deleterious effects of MCAO (line362)” and “aggravating outcomes of ischemic brain damage” (line556). Neurological scores and infarcted area, as well as oxidative stress markers, of MCAO +ATRA animals are not different if compared to MCAO group. Could you explain the basis of the reported statements?
Answer: Yes, it is true that in some cases the results of MCAO and MCAO+ ATRA are similar, or in some cases, the results of MCAO+ATRA are higher than MCAO but not significant. Even in other cases like Nrf2 quantification by ELISA and immunohistochemical analysis, the effect is significant as a lower expression of Nrf2 is observed in MCAO+ATRA than MCAO (p<0.05). ATRA seemed to have an impact greater than MCAO, though, non-significant. Based on these results, this statement was reported.
- In figure 6 Nrf2 and HO-1 appear elevated already in MCAO group compared to sham, and the treatment with A3 just increased these values. Which is the role of Nfr2 increase, beneficial or detrimental? Please comment and justify these data in the discussion.
Our findings (Figure 5) demonstrated the translocation of Nrf2 from the cytoplasm to the nucleus in MCAO group in response to the oxidative stress signals, as body’s natural defense mechanism, however, upon A3 treatment, a further upregulation of Nrf2 levels was observed, associated with higher expression of downstream HO-1, demonstrated the antioxidant potential of carveol. These effects, however, were subsided with Nrf2 inhibitor ATRA, indicating the possible role of Nrf2 in the A3 mediated neuroprotection. We have added this (561-581). All changes made are highlighted.
Furthermore, oxidative stress causes this Nrf2-Keap 1 complex to dissociates, releasing Nrf2 which translocates to the nucleus resulting in its binding to ARE and in turn regulates transcription of numerous downstream target genes as phase 2 detoxifying enzymes, antioxidant proteins, GSH generating enzymes, antioxidant genes as HO-1 and numerous others, which are involved in the antioxidative defense mechanism of the body [1]. Additionally, numerous studies have also revealed the role of Nrf2 in neuroinflammation where overexpression of Nrf2 inhibited TNF-α mediated inflammation cascade [2]. Based on these reports, we can assume that in response to the oxidative damage caused by MCAO, body’s innate Nrf2-mediated natural defense mechanism was activated resulting in increased nuclear expression of Nrf2, while treatment with A3 could further expedite this process
- Magesh, S.; Chen, Y.; Hu, L. Small molecule modulators of Keap1-Nrf2-ARE pathway as potential preventive and therapeutic agents. Med Res Rev 2012, 32, 687-726
- Kobayashi, M.; Yamamoto, M. Molecular mechanisms activating the Nrf2-Keap1 pathway of antioxidant gene regulation. Antioxidants & redox signaling 2005, 7, 385-394
- in several parameters, such as Nrf2 immunofluorescence or TLR4 in the stratum or p-NF-kB, the group of A3+MCAO+ATRA groups appear significant different from MCAO group, such as A3+MCAO animals. Could you explain these discrepancies? Please add some information in the discussion. (try to answer)
Answer: According to our study, ATRA inhibited Nrf2 signaling, but A3+ATRA+MCAO still showed a significant difference as compared to MCAO. During inflammation, normal cells expressing TLR4 significantly elevate the secretion of inflammatory mediators. TLR4 is a family of transmembrane receptors forming a part of innate immunity and is activated as a result of damaged cells expressed in neurons, astrocytes, and microglia. When they are ligated by exogenous or endogenous ligands, they trigger a proinflammatory signaling cascade thereby linking innate immunity to inflammation [1]. Based on these data, we can assume that A3 may directly inhibit the activation of TLR4 and eventually NF-kB which in turn inhibits COX-2 besides activating Nrf2 [2].
References:
- 1. Ran, S.; Bhattarai, N.; Patel, R.; Volk-Draper, L. TLR4-Induced Inflammation Is a Key Promoter of Tumor Growth, Vascularization, and Metastasis. In Translational Studies on Inflammation, IntechOpen: 2019.
- Shih, R.-H.; Wang, C.-Y.; Yang, C.-M. NF-kappaB Signaling Pathways in Neurological Inflammation: A Mini Review. Front Mol Neurosci 2015, 8, 77-77.
Nrf2 immunoflourescence—significantly higher---???
Answer: Thank you very much for raising the concern. Because it is the relative density of values, not relative integrated density in which we divide the values by sham.
Reviewer 5 Report
Alvi et al. demonstrated that an antioxidant compound A3 reduces infarct size and neurological deficits in an animal model of permanent ischemic stroke. These effects were mediated, at least in part, by Nrf2 which activates the antioxidant and anti-inflammatory responses. Importantly, ATRA, a compound that inhibits Nrf2, abolishes the protective effects of A3. Although these results are promising there are some issues that should be addressed:
Introduction:
Please rephrase this sentence: “WHO ranked stroke in the top leading causes of death and debility worldwide and which is resulted from the loss of oxygen and nutrient supply to the brain”
Please rephrase this sentence: “Ischemic stroke may either be transient or permanent and occurs as a result of an occlusion in the middle cerebral artery (MCA) and has an 80% higher incidence, compared to other types of stroke”
Methods:
Were the experimenters blinded to group allocation during surgery and behavior tests? This should be reported in the methods section.
Results:
Since the authors claim that A3 reduces neuroinflammation would be interesting to measure neutrophil infiltration and microglia activation in A3 and control group after ischemic stroke. This may be done by measuring myeloperoxidase (MPO) in brain homogenates using an ELISA kit and Iba 1 in brain sections.
Author Response
Reviewer 5:
Alvi et al. demonstrated that an antioxidant compound A3 reduces infarct size and neurological deficits in an animal model of permanent ischemic stroke. These effects were mediated, at least in part, by Nrf2 which activates the antioxidant and anti-inflammatory responses. Importantly, ATRA, a compound that inhibits Nrf2, abolishes the protective effects of A3. Although these results are promising there are some issues that should be addressed:
Introduction:
Please rephrase this sentence: “WHO ranked stroke in the top leading causes of death and debility worldwide and which is resulted from the loss of oxygen and nutrient supply to the brain”
Please rephrase this sentence: “Ischemic stroke may either be transient or permanent and occurs as a result of an occlusion in the middle cerebral artery (MCA) and has an 80% higher incidence, compared to other types of stroke”
Answer: Thank you very much, we have rephrased in the revised manuscript (line 47-49). All changes made are highlighted in the revised manuscript.
Methods:
Were the experimenters blinded to group allocation during surgery and behavior tests? This should be reported in the methods section.
Thank you very much, we have rephrased in the revised manuscript (line 184).
Results:
Since the authors claim that A3 reduces neuroinflammation would be interesting to measure neutrophil infiltration and microglia activation in A3 and control group after ischemic stroke. This may be done by measuring myeloperoxidase (MPO) in brain homogenates using an ELISA kit and Iba 1 in brain sections.
Answer: We are thankful to the respected reviewer for the constructive comments by raising some critical concerns, which will make our manuscript more valuable. We have added IHC images of microglial cell activation (Iba-1) Figure 4G (Line 377-380). All changes made are highlighted in the revised manuscript.
Round 2
Reviewer 1 Report
I really appreciate the detailed answer to my questions. Noteworthy the manuscript still contains serious pitfalls concerning data normalization.
Based on the answer given to my question 3, where I asked to normalize the obtained data by the amount of tissue, the authors indicated that the amount of cytokines measured in each sample is the result of comparison with the given absorbance and the calibration curve made with standards provided by the manufacturer. Therefore, they did not perform further relativization to the amount of sample taken for their measurements. This suggests that their results are strongly arbitrary and will strongly depend on the piece of tissue/protein that might contain more or fewer cells releasing the indicated cytokines.
Based on this answer, I am sorry, but I can’t believe that the rest of the measurements shown in this article are properly performed.
Moreover, and as previously indicated, the authors used bioinformatics tools to study the possible effect on the catalytic activity of COX-2, HO-1, p-JNK, p-NF-κB, and Nrf2. This is a very weak approximation since, in the absence of kinetic or binding data supporting these interactions, these results are not valid approximations. These results suggest a possible interaction, but they are not proved. Docking data should not be used in the absence of data that supports the docking data. I suggest the authors eliminate all the information regarding the bioinformatic analysis or to include kinetic or binding data supporting the proposed models and the interactions.
My suggestion for the authors is to correct the serious flaws and to resubmit the manuscript, in case their results are coherent and support their hypothesis.
Minor comments:
- Some of the figures are still out of the margins.
Author Response
I really appreciate the detailed answer to my questions. Noteworthy the manuscript still contains serious pitfalls concerning data normalization.
Based on the answer given to my question 3, where I asked to normalize the obtained data by the amount of tissue, the authors indicated that the amount of cytokines measured in each sample is the result of comparison with the given absorbance and the calibration curve made with standards provided by the manufacturer. Therefore, they did not perform further relativization to the amount of sample taken for their measurements. This suggests that their results are strongly arbitrary and will strongly depend on the piece of tissue/protein that might contain more or fewer cells releasing the indicated cytokines.
Answer: We are thankful to the respected reviewer for his deep and thorough reviewing of our manuscript. We appreciate the sound background knowledge and expertise of the respected reviewer. We have re-evaluated the data, also changed the methodology section (ELISA), and inserted new graphs as per new criteria by taking into account the weight of the sample.
Moreover, and as previously indicated, the authors used bioinformatics tools to study the possible effect on the catalytic activity of COX-2, HO-1, p-JNK, p-NF-κB, and Nrf2. This is a very weak approximation since, in the absence of kinetic or binding data supporting these interactions, these results are not valid approximations. These results suggest a possible interaction, but they are not proved. Docking data should not be used in the absence of data that supports the docking data. I suggest the authors eliminate all the information regarding the bioinformatic analysis or to include kinetic or binding data supporting the proposed models and the interactions.
Answer: Respected sir, as suggested, we have eliminated the docking data from our updated manuscript.
Minor comments:
- Some of the figures are still out of the margins.
We are again thankful. This time we strict to include all figures within the manuscript margin.
Thank you
Reviewer 3 Report
Authors sufficiently answered questions and explained notes and recommendations of reviewer
Author Response
Authors sufficiently answered questions and explained notes and recommendations of the reviewer.
Thank you very much.
Reviewer 4 Report
The second version of the manuscript “Post-treatment of synthetic polyphenolic 1,3,4 oxadiazole compound A3, attenuated ischemic stroke-induced neuroinflammation and neurodegeneration” appears more complete and the authors addressed most of the suggestions of the reviewer.
Nonetheless, some minor points should be considered by the author.
- In the figure 4G, the immunoreactivity for Iba1 seems to identified other cells type then microglia, that does not appear with branched phenotype, at least in sham samples. Please check if the immunohistochemistry is working properly, and add some indication for the antibody used against Iba1
- In figure 4F, 5B, 5C, 6B, 6C, 7A,7C,7D the quality of the image is still poor. Please provide, if it is possible, higher quality images.
- In figure 5C, the signals for Nrf2 (TRITC) in the images for ATRA+MCAO and A3+ATRA+MCAO samples appear weaker than the MCAO one, even if in the quantification graph (fig 5D) A3+ATRA+MCAO is significantly higher than MCAO and ATRA+MCAO display similar relative density value. The authors should provide more representative images for these two samples.
- Some minor mistake are still present in the manuscript. Please check spelling, punctuation, typographical emphasis, abbreviations through all the paper, in order to be consistent through all the paper.
Author Response
in the figure 4G, the immunoreactivity for Iba1 seems to identified other cells type then microglia, that does not appear with branched phenotype, at least in sham samples. Please check if the immunohistochemistry is working properly, and add some indication for the antibody used against Iba1
Answer: Thank you very much for reading and reviewing our manuscript which will help us to improve it to a better scientific level. We are thankful to the respected reviewer for his concern over Figure 4G. We have tried to modify the image and adds arrows for antibody identification. Moreover, in the sham group, as per our knowledge and literature review, the round cell indicates intact neuronal dendrites with a clear nucleus.
In figure 4F, 5B, 5C, 6B, 6C, 7A,7C,7D the quality of the image is still poor. Please provide, if it is possible, higher quality images.
Answer: Thank you very much for your worthy corrections We have tried to modify the resolution of Figure 5B, 5C, 6B, 6C, 7A, 7C, and 7D. We have added arrows to identify positive cells from the sham one. Please find in the revised manuscript.
In figure 5C, the signals for Nrf2 (TRITC) in the images for ATRA+MCAO and A3+ATRA+MCAO samples appear weaker than the MCAO one, even if in the quantification graph (fig 5D) A3+ATRA+MCAO is significantly higher than MCAO and ATRA+MCAO display similar relative density value. The authors should provide more representative images for these two samples.
Answer: We agree with the respected reviewer viewpoint. We, therefore, replaced and tried to improve the resolution of ATRA+MCAO and A3+ATRA+MCAO in the revised manuscript. Thank you very much
Some minor mistake are still present in the manuscript. Please check spelling, punctuation, typographical emphasis, abbreviations through all the paper, in order to be consistent through all the paper.
Answer: We are thankful to the respected reviewer for the constructive comments by raising some critical concerns, which will make our manuscript more valuable. We tried to double-check our manuscript and also by the online Grammarly software. Hope this time the reviewer will find it more refined than the previous one. Thank you
Reviewer 5 Report
In the introduction rephrase as " Stroke is the most prominent cause of human disability and may either be classified as transient or permanent and occurs as a result of an occlusion in an artery irrigating the brain. Ischemic stroke has an 80% higher incidence as compared to other types of stroke and the MCA is the most common place for the occurrence of an ischemic stroke"
Author Response
In the introduction rephrase as " Stroke is the most prominent cause of human disability and may either be classified as transient or permanent and occurs as a result of an occlusion in an artery irrigating the brain. Ischemic stroke has an 80% higher incidence as compared to other types of stroke and the MCA is the most common place for the occurrence of an ischemic stroke"
Answer: We are thankful to the respected reviewer for his deep and thorough reviewing of our manuscript. We appreciate the sound background knowledge and expertise of the respected reviewer. We have rephrased the introductory sentence as suggested by the respected reviewer. Thank you very much.